# Convergent Functions, Divergent Forms

**Hyeonseong Jeon** * [1,2]    **Ainaz Eftekhar** * [1,3]    **Aaron Walsman** [4]    **Kuo-Hao Zeng** [3]
**Ali Farhadi**[1,3]    **Ranjay Krishna**[1,3]

[1]University of Washington
[2]Seoul National University
[3]Allen Institute for AI
[4]Kempner Institute at Harvard University

**loki-codesign.github.io**

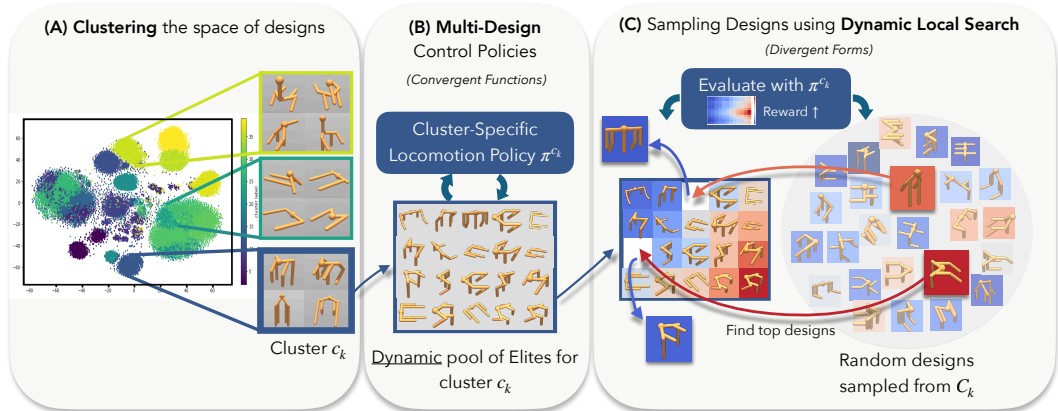

Figure 1: We introduce LOKI 🧑, a compute-efficient co-design framework that discovers diverse, high-performing robot morphologies (*divergent forms*) using shared control policies (*convergent functions*) and dynamic local search instead of mutation. (A) The design space is clustered in a learned latent space so that morphologies within each cluster share structural similarities and exhibit similar behaviors. (B) A shared control policy is trained within each cluster on a dynamic pool of elite morphologies. (C) Morphologies co-evolve with the shared policy as elites are iteratively refined through dynamic local search.

## Abstract

We introduce LOKI, a compute-efficient framework for co-designing morphologies and control policies that generalize across unseen tasks. Inspired by biological adaptation—where animals quickly adjust to morphological changes—our method overcomes the inefficiencies of traditional evolutionary and quality-diversity algorithms. We propose learning *convergent functions*: shared control policies trained across clusters of morphologically similar designs in a learned latent space, drastically reducing the training cost per design. Simultaneously, we promote *divergent forms* by replacing mutation with dynamic local search, enabling broader exploration and preventing premature convergence. The policy reuse allows us to explore $\sim 780\times$ more designs using 78% fewer simulation steps and 40% less compute per design. Local competition paired with a broader search results in a diverse set of high-performing final morphologies. Using the UNIMAL design space and a flat-terrain locomotion task, LOKI discovers a rich variety of designs—ranging from quadrupeds to crabs, bipedals, and spinners—far more diverse than those produced by prior work. These morphologies also transfer better to unseen downstream tasks

---

* Equal contribution

39th Conference on Neural Information Processing Systems (NeurIPS 2025).

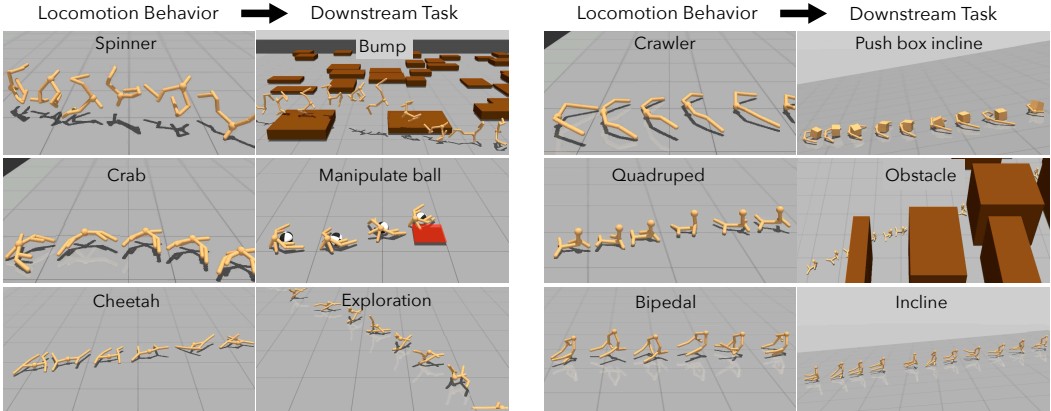

Figure 2: LOKI 🐸 discovers high-performing morphologies with **diverse locomotion behaviors** that effectively generalize across various **unseen tasks.**

in agility, stability, and manipulation domains (e.g. $2\times$ higher reward on *bump* and *push box incline* tasks). Overall, our approach produces designs that are both diverse and adaptable, with substantially greater sample efficiency than existing co-design methods.

# 1 Introduction

Embodied cognition asserts that intelligence emerges not just from the brain but is shaped by the physical embodiment [1]. This perspective acknowledges that "how a creature behaves is influenced heavily by their body," which acts as an interface between the controller and the environment [2]. A four legged embodiment *gallops* while a two legged one simply *runs*. This adaptation occurs across both evolutionary timescales—as in Darwin's finches, whose beak shapes evolved to exploit different food sources—and individual lifetimes, through mechanisms like Wolff's Law, where bones strengthen in response to load [3]. Animals can seamlessly readapt their controllers to accommodate even subtle changes in morphology [4].

Designing—or discovering—optimal embodiments for robots is a difficult combinatorial challenge [5, 6, 7, 8, 9, 10, 11, 12, 4, 13]. Co-design, the joint optimization of morphology and control, is typically framed as a bi-level optimization: an outer loop searches for morphologies (often using evolutionary strategies), while an inner loop trains a control policy for each design to evaluate its fitness (performance on a task). Due to the vastness of the design space, exhaustive search is infeasible [14, 4]. Worse, the fitness landscape is often deceptive [15], causing evolutionary strategies to converge prematurely to local optima. Escaping such traps requires maintaining diversity across the population [16, 17, 18]. Quality-diversity algorithms address this by promoting exploration of high-performing yet diverse designs [19, 20, 21, 22]. However, evaluating each candidate is costly, and diversity comes at the expense of the number of designs evaluated in the inner loop [23]. Unlike animals—who quickly adapt controllers to new bodies—robots still require retraining from scratch for each morphology, as policies transfer poorly across embodiments [4, 24].

We introduce LOKI 🐸 (**L**ocally **O**ptimized **K**inematic **I**nstantiations), an efficient framework for discovering diverse high-performing morphologies that effectively generalize across unseen tasks. Our framework is designed with the following principle: convergent functions and divergent forms. First, we mirror Wolff's Law of adapting policies to new morphologies. We train *convergent functions*, i.e. a single policy function is trained across similarly structured morphologies. We learn a latent space of possible morphologies, cluster this space and learn a shared policy that can adapt to any morphology within that cluster. These cluster-specific policies exploit shared structural features, generalizing across similar designs and behaviors, and enabling the reuse of learned behaviors. With it, we can evaluate significantly more designs without incurring additional expensive training. Competition occurs locally within each cluster-instead of a global selection pressure-resulting in diverse high-performing final designs (similar to quality-diversity algorithms but much more efficient).

Second, we break away from how most algorithms sample new designs using mutation. Mutating high-performing designs leads to limited exploration. Instead, we sample *divergent forms* by adopting dynamic local search (DLS) [25, 26, 27]. We maintain a dynamic elite pool per cluster to train a shared policy for the cluster. During training, random designs are sampled from a cluster, evaluated by the current policy, and top ones replace the worst-performing elites in the pool (see Alg. 1). Compared to mutation, which typically yields small, incremental changes, DLS enables broader and more adaptive exploration, reducing the risk of getting stuck in local minima. Since evaluations are made inexpensive with shared policies, we can afford to use random search instead of local mutation, efficiently filtering out low-quality candidates.

We use UNIMAL [4], an expressive design space encompassing approximately $10^{18}$ unique morphologies with fewer than 10 limbs. We evolve morphologies on a simple flat-terrain locomotion task. In nature, locomotion acts as a universal selection pressure across species. It is task-agnostic—helping to prevent overfitting to narrow objectives—and is efficient to simulate and easy to reward.

Beyond being substantially more efficient, LOKI discovers both genetically and behaviorally diverse high-performing morphologies, outperforming previous state-of-the-art algorithms. Using significantly less computational budget–approximately 78% fewer simulation steps (20B $\rightarrow$ 4.6B) and 40% fewer training FLOPs per design–LOKI explores $\sim 780\times$ more designs than prior evolution-based approaches (4,000 $\rightarrow$ 3M)(Table 1). LOKI discovers varied locomotion strategies including *quadrupedal, bipedal, ant-like, crab-like, cheetah, spinner, crawler*, and *rolling* behaviors (Fig. 2). In contrast, prior global competition-based evolutionary methods often converge to a single behavior type–for example, consistently producing only *cheetah-like* morphologies [4].

LOKI results in morphologies that transfer better to new unseen tasks. The final set of evolved designs are trained on a suite of downstream tasks across three domains: agility (*bump, patrol, obstacle, exploration*), stability (*incline*), and manipulation (*push box incline, manipulation ball*). Compared to baseline methods, such as DERL [4] and Map-Elites [24], LOKI's cluster-based co-evolution results in superior adaptability to these challenging unseen tasks (e.g., *bump*: 981 $\rightarrow$ 1908; *push box incline*: 1519 $\rightarrow$ 3148; see Fig. 6). Our convergent functions, shared across morphologies, results in high-performing divergent forms that generalize better to new tasks with significantly less compute.

## 2 Background and Related Work

**Brain-body co-design.** The automation of robot behavior is one of the early goals of artificial intelligence [28, 29]. In settings where the robot design is fixed, reinforcement learning (RL) [30, 31, 32, 33] has emerged as a dominant method for optimizing control policies. However, the problem is more complicated when we wish to optimize not only the robot's policy $\pi \in \Pi$, but also the its physical morphology within some design space $M \in \mathcal{M}$. This design space can be simple real-valued limb length parameters [13], more complex combinatorial structures consisting of link and joint primitives [34], or even the voxelized parameters of soft robots [35]. In these settings, the physical design of a robot and its policy are inherently coupled, as both the transition function $\mathcal{T}\left(s_{t+1} \mid s_t, a_t, M\right)$ and the total expected reward $J(\pi, M) = \mathbb{E}_{\pi, M} \sum_{t=0}^{H} \gamma^t r_t$ are now a function of both action and morphology. This means we must find ways to optimize morphology and behavior jointly: $M^* = \arg \max_M J\left(\pi_M^*, M\right)$ subject to $\pi_M^* = \arg \max_\pi \arg \max J(\pi, M)$.

Early foundational work optimized morphology and behavior jointly using genetic algorithms [14, 36]. A popular approach is to use bi-level optimization in which an inner RL loop finds the best policies for a given design, and an outer evolutionary loop selects morphologies that produce higher performing behaviors [4]. While the bi-level optimization can be computationally expensive, some authors have proposed solutions such as policy sharing with network structures that can support a variety of morphologies [5, 9]. Others discard the outer evolution loop entirely by incorporating morphology refinement [13] or compositional construction [11, 37] into the RL policy. Between these extremes are approaches that use two reinforcement learning loops that update at different schedules [6]. Recently, researchers have also attempted to produce a diverse set of morphologies rather than a single high-performing design in order to accommodate multiple downstream tasks [38, 4]. Our approach is also in this category and demonstrates substantial improvements over these prior methods.

**Optimization and Diversity.** Algorithms designed to find multiple diverse solutions to an optimization problem have a long and rich history. Island models [39, 40, 41], fitness sharing [42], tournament selection [43] and niching [44] in genetic algorithms were proposed as different ways to maintain the

diversity of a population and prevent premature collapse to a single solution mode. Similar techniques were developed for coevolutionary/multi-agent settings [45, 46].

More recently, Quality Diversity (QD) algorithms [19, 47, 21], attempt to illuminate the search space by discovering a broad repertoire of high-performing yet behaviorally distinct solutions. An important component of algorithm design in this space is deciding how to measure diversity and how to accumulate the resulting collection of solutions. Novelty Search [48, 20] measures diversity as the average distance to nearest neighbors, and adds solutions to an archive if their diversity score is higher than a threshold. MAP-Elites [23, 22] forms a discretized grid of solutions according to a user-defined feature space, and updates this map with the best solution found for each grid cell so far. This grid of solutions has been extended to Voronoi tessellations for high-dimensional feature spaces [49, 50] and combined with evolution strategies [51] and reinforcement learning [52]. QD algorithms can generate a repertoire of diverse solutions, enabling adaptive selection based on changing task demands or environmental conditions [53, 54]. In morphology optimization, evolving a diverse population on a single environment has been shown to yield better generalization and transfer to new tasks [55].

# 3 LOKI 🦝

We introduce LOKI, a compute-efficient co-design framework that discovers diverse, high-performing morphologies (*divergent forms*) using shared control policies (*convergent functions*). Leveraging structural commonalities in the design space, we train multi-design policies on clusters of similarly structured morphologies. These shared policies exploit common features, enabling behavior reuse and allowing us to evaluate many more designs without retraining. Morphologies are co-evolved alongside the policies using Dynamic Local Search (DLS) [25, 26, 27], which facilitates broader exploration within each cluster. By localizing the competition [20] within each cluster and having a broader exploration, LOKI evolves diverse high-performing solutions that effectively generalize across different unseen tasks. We describe our clustering approach in Sec.3.1, the training of cluster-specific multi-design policies in Sec.3.2, and our co-design framework in Sec. 3.3.

LOKI offers three key benefits: **Efficient Search.** We explore a design space ∼780× larger while requiring significantly fewer simulation steps (20B → 4.6B) and less compute per evaluated design (160B → 100B) (Tab. 1). **Robust Evaluation.** Single-design policies are narrowly specialized, whereas our multi-design policies are exposed to a broader range of morphologies and behaviors—leading to more robust fitness evaluations grounded in the behavioral landscape of previously encountered agents. **Diverse Solutions.** LOKI independently discovers a range of high-performing designs across clusters, without relying on explicit diversity objectives like novelty search [20].

## 3.1 Clustering the Space of Designs

Multi-design policies often struggle to generalize across morphologies with diverse behaviors. To address this, we cluster morphologies that are both structurally and behaviorally similar, and train separate policies for each cluster. Effective generalization within a cluster requires a feature space where *distance* meaningfully reflects similarity. Each morphology is represented as a sequence of limbs, with a mix of continuous and categorical parameters. Due to the complexity of this representation, clustering in the raw feature space is suboptimal. As shown in Fig. 3, K-means clustering [56] with Euclidean distance over raw parameters groups topologically distinct designs—differing overall structure—into the same cluster, despite similarities in simpler attributes like limb length or radius.

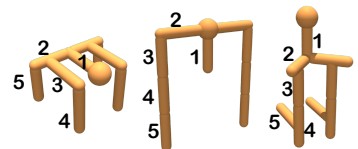

Figure 3: **Clustering using raw morphology parameters fails to capture the topological similarity**. The 3 designs are from the same cluster. Indices show the DFS traversal.

Instead, we train a Transformer-based VAE [57, 58] to encode morphologies into a structured latent space. Each morphology is represented as an $L \times D$ matrix, where $L$ is the maximum number of limbs and $D = 47$ is the number of parameters per limb, with both continuous (e.g., *limb orientation, length, joint gear*) and categorical (e.g., *joint type, joint angle, depth*) attributes. Limb vectors are ordered using a depth-first search (DFS) traversal [59], starting from the head, which encodes *torso type (horizontal/vertical), head density*, and *radius*. If a morphology has fewer than $L$ limbs, the sequence is zero-padded. The transformer encoder [57] maps each morphology to a $(L \times H)$ latent

Table 1: **Efficiency and Coverage Comparison.** LOKI explores a much larger design space while requiring significantly fewer simulation steps and less training FLOPs per evaluated design See Appendix F for details.

|  | # Interactions (B) $\downarrow$ | # Searched Morphologies $\uparrow$ | FLOPs per Morphology (B) $\downarrow$ |
|---|---|---|---|
| DERL | 20 | 4000 | 160 |
| LOKI | **4.62** | **3,120,000** | **100** |

space, where $H < D$. To support decoding, we append an EOS (end-of-sequence) token to mark the final limb and use depth tokens to indicate each limb's position in the depth-first traversal.

The loss over a batch is defined as $\mathcal{L}^{\text{recon}} = \sum_{i=1}^{L} \mathcal{L}_i^{\text{recon}}$, where $\mathcal{L}_i^{\text{recon}} = \frac{\sum_j \ell_{\text{cont}}(\hat{y}_{ij}, y_{ij}) \cdot \mathcal{M}_{ij}^{\text{cont}}}{\sum_j \mathcal{M}_{ij}^{\text{cont}}} +$ $\frac{\sum_j \ell_{\text{cat}}(\hat{y}_{ij}, y_{ij}) \cdot \mathcal{M}_{ij}^{\text{cat}}}{\sum_j \mathcal{M}_{ij}^{\text{cat}}}$ is the reconstruction loss for the $i$-th limb. $\ell_{\text{cont}}$ and $\ell_{\text{cat}}$ are the L2 loss and cross-entropy loss for continuous or categorical attributes. $\mathcal{M}_{ij}^{\text{cont}}$ and $\mathcal{M}_{ij}^{\text{cat}}$ are binary masks showing existence of the $j$-th attribute in the $i$-th limb, and $y_{ij}$ is the ground-truth value. Instead of the standard ELBO objective, we adopt an adaptive scheduling strategy for the KL divergence scaling factor $\beta$ [60]. Our goal is to learn a compact latent representation suitable for clustering. Therefore, we prioritize minimizing the reconstruction loss over enforcing strong regularization towards a standard normal prior. Following [61], we exponentially decay $\beta$ within a predefined range $[\beta_{\text{min}}, \beta_{\text{max}}]$.

We generate 500k unique random morphologies within the UNIMAL space [4] (see Appendix C for details). We train a transformer-based VAE (4 layers, 4 heads, latent dimension $H = 32$) on these designs using a batch size of 4096, an initial learning rate of $10^{-4}$, and a single A40 GPU for 200 epochs. We then cluster the 500k morphologies into $N_c = 40$ groups using the K-means algorithm [56] based on Euclidean distance of their corresponding VAE latent codes. A t-SNE [62] visualization of the clusters is shown in Fig. 1-A.

## 3.2 Convergent Functions: Multi-Design Control Policies

Most prior evolution-based co-design approaches train each morphology from *scratch* using reinforcement learning, without sharing experience across the population. In contrast, we leverage structural commonalities within the design space by training multi-design policies for clusters of morphologies that share structural and behavioral similarities. These cluster-specific policies serve as surrogate scoring functions, enabling efficient evaluation of a large number of designs within each cluster without retraining, thereby significantly improving **search efficiency**.

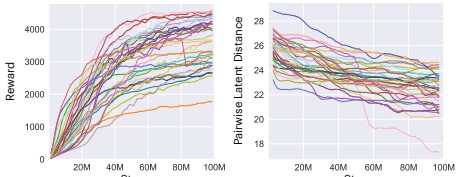

Figure 4: **Coevolution of morphologies with multi-design policies.** *Left*: Training rewards for each policy. *Right*: Mean pairwise distance of morphologies in the elite pool of each cluster.

As shown in Fig. 1-B, for each cluster, we initialize a training pool $\mathcal{P}_0^{C_k}$ for $C_k$ with $N_w = 20$ randomly sampled agents. We train a multi-design policy $\pi_\theta^{C_k}$ from scratch using the dynamic agent pool within that cluster for $N_{\text{iter}} = 1220$ training iterations. Every $f_{\text{diff}} = 2$ iterations, the pool is updated $\mathcal{P}_i^{C_k} \to \mathcal{P}_{i+1}^{C_k}$, to identify the cluster's elites via the stochastic search process described in Sec. 3.3. Each cluster-specific policy is trained on a large number of designs, enabling generalization across similar morphologies and behaviors.

We adopt the Transformer-based policy architecture (5 layers/2 heads) from [63] for our multi-design policies. Each policy is trained for 100 million simulation steps using a batch size of 5120 and an initial learning rate of $3 \times 10^{-4}$ with cosine annealing. Training is distributed across six A40 GPUs, with each GPU handling 6–7 cluster-specific policies in parallel (more details in the Appendix H).

## 3.3 Divergent Forms: Co-evolving Elites using Dynamic Local Search within Clusters

Most evolution-based approaches generate new designs by mutating existing ones. However, such *small incremental changes* often restrict exploration to local neighborhoods, increasing the risk of premature convergence to local optima. Additionally, *global selection pressure* can cause the search to favor a single dominant species or behavior—often biasing toward short-term gains like faster

---

**Algorithm 1 : LOKI** 🧌

---

1: **Input:** Morphology clusters $\mathcal{C} = \{\mathcal{C}_k\}_{k=1}^{N_c}$, training iterations $N_{\text{iter}}$, initial pool size $N_w$,
    update frequency $f_{\text{diff}}$, sample size $N_{\text{sample}}$, replace count $N_{\text{filter}}$
2: **for** each cluster $\mathcal{C}_k \in \mathcal{C}$ **do**
3:     Initialize training pool $\mathcal{P}_0^{\mathcal{C}_k}$ with $N_w$ random morphologies from $\mathcal{C}_k$
4:     Initialize policy $\pi_\theta^{\mathcal{C}_k}$ and value function $V_\theta^{\mathcal{C}_k}$ from scratch
5:     **for** $i = 1$ to $N_{\text{iter}}$ **do**
6:         **if** $i \bmod f_{\text{diff}} == 0$ **then**
7:             Sample $N_{\text{sample}}$ new morphologies from $\mathcal{C}_k$ and evaluate using $\pi_\theta^{\mathcal{C}_k}$
8:             Select top $N_{\text{filter}}$ sampled morphologies $\{\hat{w}_j\}_{j=1}^{N_{\text{filter}}}$
9:             Identify $N_{\text{filter}}$ lowest-reward agents $\{w_j\}_{j=1}^{N_{\text{filter}}}$ in $\mathcal{P}_{i-1}^{\mathcal{C}_k}$
10:            Replace: $\mathcal{P}_i^{\mathcal{C}_k} \leftarrow \mathcal{P}_{i-1}^{\mathcal{C}_k} \setminus \{w_j\} \cup \{\hat{w}_j\}$
11:         **else**
12:            $\mathcal{P}_i^{\mathcal{C}_k} \leftarrow \mathcal{P}_{i-1}^{\mathcal{C}_k}$
13:         **end if**
14:         PPO_step($\pi_\theta^{\mathcal{C}_k}, V_\theta^{\mathcal{C}_k}, \mathcal{P}_i^{\mathcal{C}_k}$)
15:     **end for**
16: **end for**

---

learning, while overlooking more complex or unconventional strategies (e.g., spinning) that require longer-term training (Fig. 2).

To overcome these limitations, we sample *divergent forms* using Dynamic Local Search (DLS) [25, 26, 27], combined with localized competition within each cluster. This allows clusters to independently evolve high-performing and behaviorally **diverse** morphologies (see different locomotion behaviors in Fig. 2). DLS promotes broader and more adaptive exploration within each cluster, reducing the risk of getting stuck in local minima. Our multi-design policy $\pi_\theta^{C_k}$ acts as a dynamic evaluation function, continually updated via PPO [32] on a dynamic pool of elite designs $\mathcal{P}_i^{C_k}$. This enables efficient assessment of diverse candidates sampled *broadly* across the cluster, rather than relying on narrowly mutated variants. Because evaluations are made efficient through policy sharing, we can use random search instead of mutation—allowing us to quickly filter out poor designs and retain promising ones.

As shown in Fig. 1-C, every $f_{\text{diff}} = 2$ training iterations, $N_{\text{sample}} = 128$ new random designs are sampled from the cluster and evaluated using the current policy $\pi_\theta^{C_k}$. The top $N_{\text{filter}} = 2$ designs replace the lowest-performing $N_{\text{filter}}$ agents in the pool. Training then continues with the updated pool $\mathcal{P}_{i+1}^{C_k}$. Fig. 4 shows the training rewards and pairwise distances of the designs within each training pool $\mathcal{P}_{C_k}$. Over time, each pool converges to a steady state, where the same $N_{\text{filter}} = 2$ agents are consistently replaced in each update cycle. The complete algorithm is shown in Alg. 1.

**How to choose the number of clusters?** Number of clusters plays a key role in balancing intra-cluster behavioral similarity with the breadth of localized competition. Within each cluster, morphologies should be behaviorally similar enough for the policy to generalize effectively, while the clustering should remain fine-grained enough to capture morphological diversity. With $N_c = 10$, clusters roughly align with limb count, resulting in overly coarse groupings. We select $N_c = 40$ to better capture finer morphological distinctions while maintaining a feasible computational budget. Each policy evaluates up to $\left\lfloor \frac{N_{\text{iter}}}{f_{\text{diff}}} \right\rfloor \cdot N_{\text{sample}} + N_w \sim 78{,}000$ samples during training, allowing each morphology to be evaluated approximately 6 times on average. These repeated evaluations are valuable, as early policies may be unreliable; re-evaluation with improved policies can uncover better-performing behaviors (see Appendix for details).

## 4 Experiments

In all experiments, morphologies are evolved for locomotion on flat terrain (FT) using UNIMAL space. Locomotion is a universal and ubiquitous evolutionary pressure across species; it is task-agnostic, avoids overfitting to narrow objectives, and is easy to simulate and reward.

We evaluate both the *performance* and *diversity* of the final evolved morphologies, comparing them to other evolution-based co-design methods as well as Quality-Diversity approaches (Sec. 4.1). LOKI

Table 2: LOKI maintains both **quality** and **diversity**. It obtains the highest QD-Score, demonstrating its ability to find high-performing solutions across a wide range of niches. ($\pm$ denotes standard error across 4 training seeds)

| Method | Max Fitness | QD-score | Coverage(%) |
|---|---|---|---|
| RANDOM | $3418.9 \pm 390.8$ | 32.1 | 90 |
| DERL | $\mathbf{5760.8} \pm 248.2$ | 26.2 | 37.5 |
| MAP-ELITES | $\mathbf{5807.3} \pm 196.5$ | 43.5 | 65.0 |
| LATENT-MAP-ELITES | $5257.0 \pm 491.3$ | 38.1 | **100** |
| LOKI(W/O CLUSTER) | $4825.8 \pm 76.1$ | 40.0 | 75 |
| LOKI ($N_c = 10$) | $4008.0 \pm 363.7$ | 21.6 | 47.5 |
| LOKI ($N_c = 20$) | $5544.0 \pm 268.1$ | 26.6 | 45.0 |
| **LOKI** 🦎 ($N_c = 40$) | $\mathbf{5671.9} \pm 360.1$ | **60.9** | **100** |

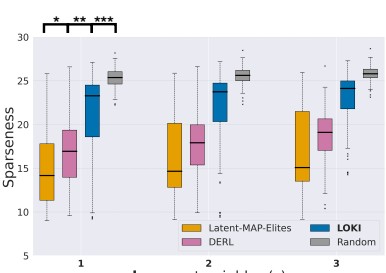

Figure 5: LOKI evolves significantly more **sparse** solutions (measured by the average distance to $k$-nearest neighbors). This is due to using Dynamic Local Search for sampling designs, rather than mutations. Higher sparseness indicates more diversity. (Statistical significance was assessed using independent samples t-tests; $*P < 0.005$; $**P < 10^{-11}$; $***P < 10^{-16}$).

promotes both genetic and behavioral diversity, discovering a wide range of locomotion strategies including *quadrupedal, bipedal, crab-like, cheetah, spinner, crawler*, and *rolling* behaviors, as shown in Fig. 2). We assess the transferability of the evolved morphologies to a suite of downstream tasks across three domains: agility (*patrol, obstacle, bump, exploration*), stability (*incline*), and manipulation (*push box incline, manipulation ball*). Compared to baselines, LOKI exhibits superior adaptability to these unseen challenges—at both the morphology and policy levels—driven by the quality and diversity of the evolved designs (Sec. 4.2).

**Baselines and Experiment Details.** We compare our method with DERL [4], a prior evolution-based co-design approach, and CVT-Map-Elites [22], a representative Quality-Diversity baseline. For a fair comparison, we evaluate the top $N = 100$ final evolved morphologies (elites) from each algorithm. For methods using $N_c = 40$ clusters, we select the top 2–3 agents from the final training pool of each cluster to form the final population. We include the following baselines and ablations:

1. RANDOM: 100 morphologies are randomly sampled from the design space.

2. DERL [4]: We use the publicly released set of $N = 100$ agents evolved on flat terrain. DERL introduces a scalable framework for evolving morphologies in the UNIMAL design space using a bi-level optimization strategy: an outer evolutionary loop mutates designs, while an inner reinforcement learning loop trains a separate MLP policy for each agent for over 5 million steps. DERL employs distributed, asynchronous evolutionary search to parallelize this process, evolving and training 4,000 morphologies over an average of 10 generations to select the top 100.

3. MAP-ELITES [23, 49]: A Quality-Diversity (QD) baseline where the morphology space is discretized into $N_c = 40$ niches using K-means clustering on raw morphology parameters. Each cluster serves as a cell in the MAP-Elites archive maintaining 3 elites throughout evolution. The algorithm explores the same number of morphologies as DERL (40 generations $\times$ 100 morphologies per generation=4000), training a separate MLP policy for each agent for 5M steps.

4. LATENT-MAP-ELITES: Similar to MAP-ELITES, but morphology space is discretized into $N_c = 40$ niches using K-means clustering in the VAE latent space rather than raw parameters.

5. **LOKI** (W/O CLUSTER): An ablation of our method with no clustering. Morphologies are evolved over 40 independent runs, each initialized with a pool of $N_w$ agents sampled randomly from the full design space rather than within clusters.

6. **LOKI**: Our full co-design framework using $N_c = 40$ clusters in the VAE latent space.

## 4.1  Performance and Diversity of the Evolved Designs

The design space includes both *fast* and *slow* learning morphologies, requiring varying amounts of interaction to master locomotion on flat terrain. Training single-design controllers from scratch is constrained by limited budgets—e.g. DERL [4] trains each morphology using MLP policies

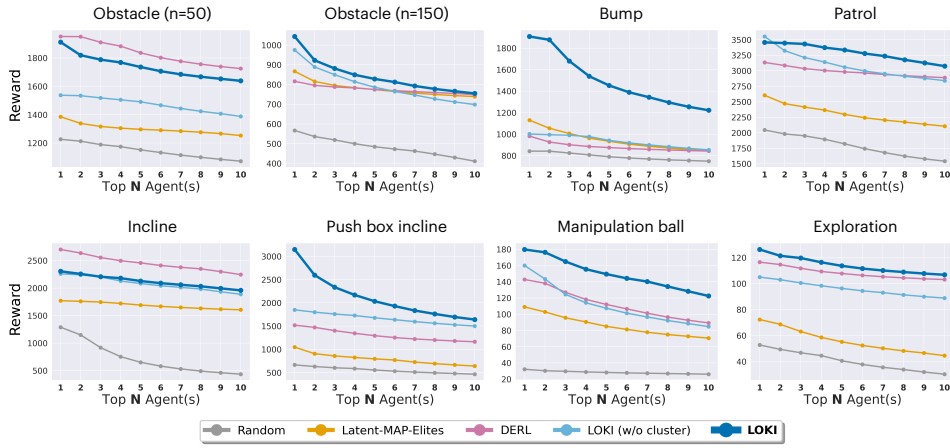

Figure 6: **Morphology-level task adaptability.** We evolve diverse morphologies on flat terrain that generalize more effectively to unseen tasks requiring varied skills. DERL morphologies are overfitted to flat terrain and perform best on *obstacle (n=50)* and *incline*, which are structurally similar. LOKI shows significantly better adaptability on *bump* ($981 \rightarrow 1908$), *push box incline* ($1519 \rightarrow 3148$), *manipulation ball* ($142 \rightarrow 172$) enabled by its morphological diversity (e.g., crawlers, crabs) and the emergence of more complex behaviors (e.g., spinning, rolling).

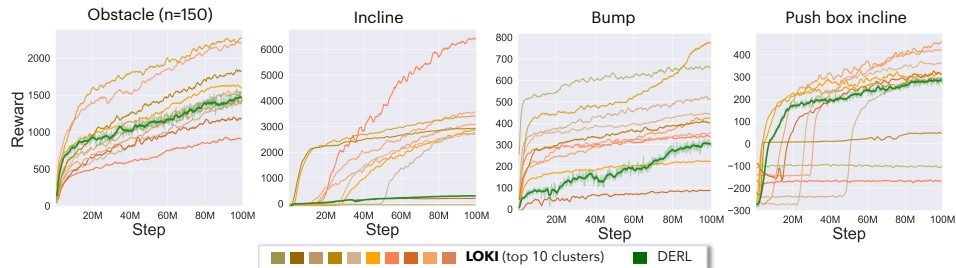

Figure 7: **Policy-level task adaptability.** Our co-evolution framework not only produces a diverse set of morphologies but also cluster-specific policies that generalize more effectively to unseen tasks. Each cluster captures distinct morphologies and behaviors, enabling its policy to better adapt to tasks aligned with those traits.

for 5 million steps, which is often insufficient to learn more complex behaviors, such as spinning (see Appendix B). Global competition causes the algorithm to favor faster learners like *cheetah-like* morphologies, reducing the diversity of behaviors. In contrast, LOKI promotes both *fast* and *slow* learners by localizing competition within clusters, using powerful cluster-specific policies, and incorporating Dynamic Local Search (DLS) to encourage broader exploration within each cluster. This leads to greater behavioral and morphological diversity (Fig. 2).

In Tab. 2, we compare LOKI for locomotion on flat terrain (FT) against baselines using three core metrics commonly used to evaluate Quality-Diversity (QD) algorithms [64, 19]: *Maximum Fitness* (the highest reward achieved by any agent in the final population), *Coverage* (the number of clusters filled by the final population), and *QD-Score* (sum of the highest fitness values discovered in each cluster, capturing both quality and diversity). For fair comparison of QD-score and coverage, we assign top 100 agents from each baseline to $N_c$=40 latent clusters (details in Appendix E).

While LOKI achieves slightly lower maximum fitness than DERL and MAP-ELITES, it obtains the highest QD-Score, demonstrating its ability to find high-performing solutions across a wide range of ecological niches. Although LATENT-MAP-ELITES achieves full coverage by evolving agents within the same latent clusters, it struggles to discover high-quality solutions in many of them, resulting in a significantly lower QD-Score. DERL, on the other hand, is optimized solely for fitness, not

diversity. As a result, it overfits to the training task (flat terrain locomotion) and lacks behavioral variety—reflected in its low coverage score of 37.5%, with only 15 out of 40 clusters filled.

Our ablations further highlight the importance of clustering. When clusters are too large and contain morphologies with different behavior types, the shared policy struggles to generalize effectively, which explains the performance drop at $N_c = 10$ and $N_c = 20$.

Fig. 5 compares the *sparseness* [20] of solutions as a novelty metric. A simple measure of sparseness $\rho$ at a point $x$ is $\rho(x) = \frac{1}{k} \sum_{i=1}^{k} \text{dist}(x, \mu_i)$, where $\mu_i$ is the $i$th-nearest neighbor of $x$ in the morphology latent space. LOKI achieves the highest sparseness among the three learning-based methods, closely approaching RANDOM. This is largely due to LOKI's use of Dynamic Local Search (DLS) to sample diverse designs within each cluster, as opposed to the incremental mutations used in DERL and LATENT-MAP-ELITES, which tend to produce solutions in local neighborhoods.

## 4.2 Transitioning to New Tasks and Environments

**Morphology-Level.** LOKI produces a diverse set of solutions that are better suited for adaptation to various unseen tasks and environments, reducing the need to re-evolve agents for each setting. We evaluate generalization across eight test tasks, each requiring a distinct set of skills. For each method, the final set of $N = 100$ evolved morphologies (elites) is independently trained from scratch on each test task using MLP-based policies, with 5 random seeds and training durations of 5, 15, or 20 million steps depending on task difficulty. *Obstacle (n=150)* is more challenging with three times more obstacles than *obstacle (n=50)*. In the *bump* task, agents must learn behaviors like jumping or climbing without explicit information about the bump's location. All tasks except *obstacle (n=150)* and *bump* were originally introduced in [4] (more details in Appendix D).

Fig. 6 shows the cumulative mean reward of the top 10 agents from each method on each task. LOKI outperforms the DERL baseline on 6 out of 8 tasks, indicating that the diverse set of morphologies evolved for flat terrain transfers more effectively to new environments. In contrast, DERL agents exhibit limited behavioral diversity and are overfitted to the training task—they perform best on *obstacle (n=50)* and *incline*, likely because these tasks are structurally similar to flat terrain. LOKI achieves large gains on tasks that differ more significantly from locomotion, such as *bump* ($981 \rightarrow 1908$), *push box incline* ($1519 \rightarrow 3148$), and *manipulate ball* ($142 \rightarrow 172$). These improvements stem from the emergence of diverse morphologies (e.g., crawlers, crabs) and complex behaviors (e.g., spinning, rolling) that DERL and MAP-ELITES fail to discover (see Fig. 2).

**Policy-Level.** Each multi-design policy is trained on a large number of morphologies within its cluster using a dynamic pool that is updated with new designs every few iterations. We evaluate the generalization capabilities of these policies on four new tasks. Since each cluster captures a distinct subset of morphologies and behaviors, the corresponding policy is expected to adapt more effectively to tasks aligned with those traits.

We fine-tune the top 10 cluster-specific policies—pretrained on flat-terrain locomotion—on each new task. As a baseline, we pretrain the Transformer policy architecture on the top $N_w = 20$ morphologies evolved by DERL on flat terrain and fine-tune it similarly. Fig. 7 shows that for each task, certain cluster-specific policies adapt significantly better after fine-tuning, as their corresponding morphologies are inherently aligned with the skills required. This highlights the benefit of co-evolving diverse morphologies alongside specialized controllers within structurally coherent clusters. The intra-cluster similarity further aids generalization, enabling policies to learn transferable behaviors. Overall, our framework not only produces diverse morphologies but also yields cluster-specific policies that generalize more effectively to unseen tasks.

## 5 Conclusion

We introduce LOKI, a compute-efficient co-design framework that discovers diverse, high-performing robot morphologies (*divergent forms*) using shared control policies (*convergent functions*). By clustering structurally similar morphologies, we train multi-design policies that enable behavior reuse and allow for the evaluation of significantly more designs without retraining. Morphologies are co-evolved with policies using Dynamic Local Search (DLS) rather than incremental mutations, allowing broader exploration within each cluster. Localized competition combined with broader exploration

leads LOKI to outperform Quality Diversity (QD) algorithms in evolving diverse locomotion strategies. Our agents also generalize better to unseen tasks compared to prior evolution-based and QD methods.

## 6  Limitations

One limitation of our method is that the number of clusters—and consequently, the diversity of discovered solutions—must be determined at the start of training. While this could be partially addressed by adding new morphology clusters later to better cover poorly performing regions, doing so may require costly retraining. Additionally, training costs scale linearly with the number of clusters. In future work, this limitation could be alleviated by introducing an adaptive mechanism that dynamically grows or shrinks the number of clusters based on their performance during training.

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

## Appendices for *Convergent Functions, Divergent Forms*

The following items are provided in the Appendix:

- *Benefits of clustering the design space in the **learned latent space** instead of using raw morphology parameters (App.A)*
- *Analysis of the locomotion **learning speed** of our evolved agents (fast and slow learners) (App.B)*
- *Procedure for **sampling random morphologies** within the UNIMAL space (App.C)*
- *Detailed description of the different **test tasks** (App.D)*
- *Detailed description of the **QD-score** metric (App.E)*
- *Description of how **efficiency metrics** in Tab.1 are calculated (App.F)*
- *Pseudo-code for the **Map-Elites** baseline (App.G)*
- *Full **hyperparameter** details for the experiments (App. H)*
- *Broader Societal Impacts (App. I)*

On our website (loki-codesign.github.io), we have

- *Videos showing the diverse locomotion behaviors of* LOKI*'s evolved agents*
- *Videos of the evolved agents transferred to new test tasks (Obstacle, Bump, Incline, Push box incline, Manipulation ball, Exploration)*

## A    Benefits of Clustering in the Learned Latent Space

To investigate the role of latent-space clustering versus clustering based on raw morphology parameters, we conduct an ablation study: LOKI (W/ RAW-PARAM-CLUSTER), a variant of our method in which latent-space clusters are replaced with clusters computed from raw morphology parameters, while keeping the number of clusters fixed at $N_c = 40$.

Fig.8 overlays the cumulative mean reward of the top 10 agents from LOKI (W/ RAW-PARAM-CLUSTER) and MAP-ELITES (W/ RAW-PARAM-CLUSTER) on top of Fig.6 in the main paper. Notably, LOKI outperforms LOKI (W/ RAW-PARAM-CLUSTER) across all tasks, indicating that our co-evolution framework benefits from structuring the morphology space via a learned latent representation. In contrast, clustering in the raw parameter space fails to capture structural and behavioral similarities, leading to poor generalization of multi-design policies within each cluster. Tab. 3 also compares LOKI against LOKI (W/ RAW-PARAM-CLUSTER) across the quality-diversity metrics.

## B    Learning Speed of the Evolved Morphologies (Fast & Slow Learners)

We examine the correlation between learning speed and the final performance of evolved agents. To evaluate performance across different training durations, we train single-agent MLP policies [4] for both 5M and 15M steps on flat terrain. This evaluation excludes agents from clusters with consistently low training rewards or predominantly simple morphologies. As shown in Fig. 9, the top-performing agents under short- and long-term training are largely disjoint, revealing the presence of both *fast-* and *slow*-learning morphologies. Notably, high short-term performance does not necessarily indicate high long-term performance—some slow learners achieve superior results after 15M steps despite underperforming at 5M steps.

Prior approaches such as DERL train single-agent policies for a fixed number of steps (e.g., 5M), which biases evolution toward fast learners—often resulting in morphologies dominated by cheetah-like forms. In contrast, LOKI does not rely on short-term training. It leverages multi-design transformer policies trained over significantly longer durations, enabling the discovery of both fast and slow learners across the morphology clusters.

Table 3: LOKI benefits both **quality** and **diversity** from clustering in a structured morphology latent space. *One new baseline,* LOKI (W/ RAW-PARAM-CLUSTER), marked with (*), is added to Tab. 2 to assess the impact of clustering in raw parameter space. (± denotes standard error across 4 training seeds.)

| Method | Max Fitness | QD-score | Coverage(%) |
|---|---|---|---|
| RANDOM | $3418.9 \pm {}_{390.8}$ | 32.1 | 90 |
| DERL | $\mathbf{5760.8} \pm {}_{248.2}$ | 26.2 | 37.5 |
| MAP-ELITES (W/ RAW-PARAM-CLUSTER) | $\mathbf{5807.3} \pm {}_{196.5}$ | 43.5 | 65.0 |
| LATENT-MAP-ELITES | $5257.0 \pm {}_{491.3}$ | 38.1 | **100** |
| LOKI (W/O CLUSTER) | $4825.8 \pm {}_{76.1}$ | 40.0 | 75 |
| LOKI ($N_c = 10$) | $4008.0 \pm {}_{363.7}$ | 21.6 | 47.5 |
| LOKI ($N_c = 20$) | $5544.0 \pm {}_{268.1}$ | 26.6 | 45.0 |
| LOKI (W/ RAW-PARAM-CLUSTER)$^{(*)}$ | $4055.0 \pm {}_{323.7}$ | 27.5 | 62.5 |
| **LOKI** ($N_c = 40$) | $\mathbf{5671.9} \pm {}_{360.1}$ | **60.9** | **100** |

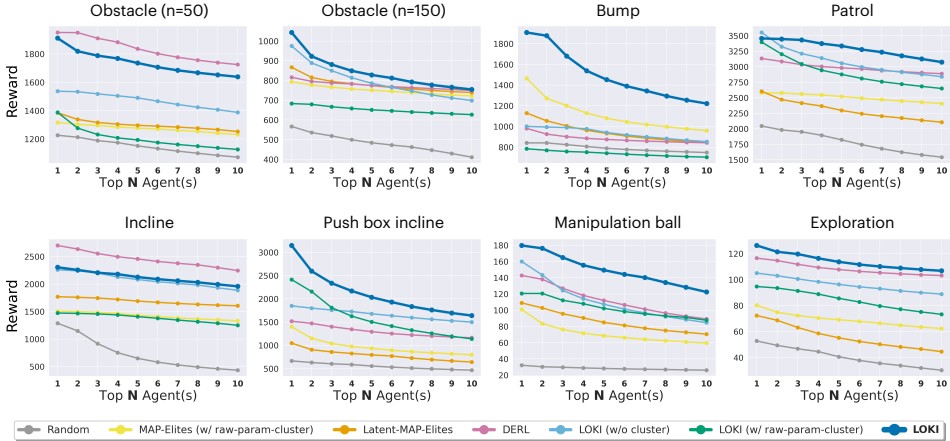

Figure 8: **Morphology-level task adaptability.** Two baselines (MAP-ELITES (W/ RAW-PARAM-CLUSTER), LOKI (W/ RAW-PARAM-CLUSTER)) are added to Fig. 6 to assess the impact of clustering in raw parameter space.

## C  Sampling Random Morphologies within the UNIMAL Space

The process of sampling random morphologies follows the procedure introduced in prior work [4]. During the population initialization phase, a new morphology is created by first sampling the total number of limbs to grow, followed by a series of mutation operations until the desired number of limbs is reached. These mutations include: *growing or deleting limbs*, *mutating limb parameters, density, degrees of freedom (DoF), gear ratios, and joint angles*. For each mutation, the parameters are uniformly sampled from predefined ranges specified in [4]. The key difference is that DERL [4] samples parameter values from discrete sets, while LOKI samples continuously within each range, enabling a denser and more comprehensive coverage of the morphology space. Table 4 presents the parameter ranges used to create our morphologies. The notation *range(a, b)* denotes a continuous range from *a* to *b*.

## D  Test Task Descriptions

We describe the task specifications and required skills for the eight test tasks used in our morphology-level evaluation. All tasks, except for *Bump* and *Obstacle (n=150)*, were introduced in prior work [4]. We adopt the same task configurations and reward structures as outlined therein.

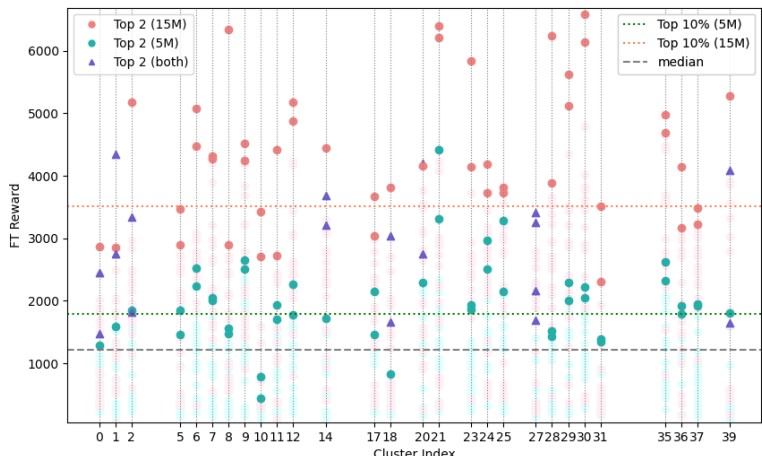

Figure 9: **Training rewards across short- and long-term learning.** Final rewards for LOKI's agents after 5M and 15M training steps. The top-performing agents at short and long training horizons are largely disjoint. High short-term performance does not guarantee long-term success—some slow learners significantly outperform fast learners after extended training.

| Parameters | Sampling Range |
|---|---|
| Max limbs | 11 |
| Limb radius | range(0.02, 0.06) |
| Limb height | range(0.2, 0.4) |
| Limb density | range(500, 1000) |
| Limb orientation $\theta$ | [0, 45, 90, 135, 180, 225, 270, 315] |
| Limb orientation $\phi$ | [90, 135, 180] |
| Head radius | 0.10 |
| Head density | range(500, 1000) |
| Joint axis | [x, y, xy] |
| Motor gear range | range(150, 300) |
| Joint limits | [(-30, 0), (0, 30), (-30, 30), (-45, 45), (-45, 0), (0, 45), |
| | (0, -60), (0, 60), (-60, 60), (-90, 0), (0, 90), (-60, 30), (-30, 60)] |

Table 4: Design parameters used for sampling random morphologies in the UNIMAL space.

**Patrol.** The agent must repeatedly traverse between two target points separated by 10m along the x-axis. High performance in this task requires rapid acceleration, short bursts of speed, and quick directional changes. (Training steps: 5 million)

**Incline.** The agent operates in a $150 \times 40\text{m}^2$ rectangular arena inclined at a 10-degree angle. The agent is rewarded for moving forward along the +x axis. (Training steps: 5 million)

**Push Box Incline.** The agent must push a box with a side length of 0.2m up an inclined plane. The environment is a $80 \times 40\text{m}^2$ rectangular arena tilted at a 10-degree angle. The agent starts at one end of the arena and is tasked with propelling the box forward along the slope. (Training steps: 5 million)

**Obstacle (n=50, 150).** The agent must navigate through a cluttered environment filled with static obstacles to reach the end of the arena. Each box-shaped obstacle has a width and length between 0.5m and 3 m, with a fixed height of 2 m. *n* denotes the number of randomly distributed obstacles across a flat $150 \times 60\text{m}^2$ terrain. (Training steps: 5 million)

**Bump.** The agent must traverse an arena filled with 250 low-profile obstacles randomly placed on a flat $150 \times 60\text{m}^2$ terrain. Each obstacle has a width and length between 0.8m and 1.6m, and a height between 0.1m and 0.25m. As obstacle height is comparable to the agent's body, this task promotes behaviors such as jumping or climbing, adding complexity to locomotion and body coordination. (Training steps: 15 million)

**Manipulate Ball.** The agent must move a ball from a random source location to a fixed target. A ball of radius 0.2 m is placed at a random location in a flat, square $30 \times 30\text{m}^2$ arena, with the agent initialized at the center. This task requires a fine interplay of locomotion and object manipulation, as the agent must influence the ball's motion through contact while maintaining its own balance and stability. (Training steps: 20 million)

**Exploration.** The agent begins at the center of a flat $100 \times 100\text{m}^2$ arena divided into $1 \times 1\text{m}^2$ grid cells. The goal is to maximize the number of unique grid cells visited during an episode. Unlike previous tasks with dense locomotion rewards, this task provides a sparse reward signal. (Training steps: 20 million)

## E  QD-Score (a quality-diversity metric)

QD-score [64, 19, 55] is a more comprehensive metric than maximum fitness, as it captures not only the performance of the single best agent but also the diversity of high-performing solutions across the search space. Given the vast combinatorial complexity of the UNIMAL space, QD-score is particularly well-suited for evaluating the quality and spread of evolved morphologies.

In Tab. 2, we report the percentile QD-score computed over $N_c = 40$ latent morphology clusters, defined as:

$$\text{QD-score} = \frac{1}{N_c} \sum_{i=1}^{N_c} \mathbb{1}_{\|\mathcal{M}_i\|>0} \cdot \frac{f_i - f_{\min}}{f_{\max} - f_{\min}} \tag{1}$$

Here, $\mathcal{M}_i$ denotes the set of evolved morphologies allocated to the $i$-th cluster, and $f_i$ is the mean fitness of the best-performing agent in that cluster. $f_{\max}$ and $f_{\min}$ represent the maximum and minimum mean fitness values across all final population clusters, respectively. This normalization ensures comparability across clusters with different fitness scales.

## F  Efficiency Metrics in Table 1.

This section defines the efficiency metrics reported in Tab. 1.

**Number of interactions.** The number of interactions refers to the total number of environment steps taken by all agents throughout the entire evolutionary process.

In our multi-design evolution framework, two types of interactions are counted: (1) trajectories collected from agents in the training pool, which are used to train the shared multi-design policy via PPO, and (2) evaluation episodes conducted during each drop-off round.

The total number of interactions is calculated as:

$$\text{Total interactions} = N_c \cdot (\text{\# interactions }_{\text{(per cluster)}} + \text{\# evaluated samples} \cdot \text{episode length})$$
$$= N_c \cdot \left( 100\text{M} + \left\lfloor \frac{N_{\text{iter}}}{f_{\text{diff}}} \right\rfloor \cdot N_{\text{sample}} \cdot l_{\text{eval}} \right)$$
$$\approx 4.62\text{B} \tag{2}$$

In contrast, DERL trains 4,000 agents independently using a single-agent MLP policy, with each agent trained for 5M environment steps—resulting in a total of 20B interactions. Since DERL uses the training reward directly for agent selection, no separate evaluation phase is required.

**Number of searched morphologies.** The total number of morphologies explored across the morphology space is calculated as follows:

$$\text{\# of covered morphologies }_{\text{(per cluster)}} \leq \left\lfloor \frac{N_{\text{iter}}}{f_{\text{diff}}} \right\rfloor \cdot N_{\text{sample}} + N_w \approx 78{,}000 \tag{3}$$

$$\text{\# of searched morphologies} = \text{\# of covered morphologies}_{\text{(per cluster)}} \cdot N_c$$
$$\approx 78{,}000 \cdot 40$$
$$= 3.12\text{M} \qquad (4)$$

Note that each morphology is evaluated approximately 6 times on average. These repeated evaluations are valuable because early-stage policies may be unreliable. Re-evaluating morphologies with progressively improved policies allows for the discovery of higher-performing behaviors.

**FLOPs per searched morphology.** Given the MLP and Transformer policy model architectures used in prior work [4, 63], we compute the training FLOPs per searched morphology as shown in Eq. 5 with the values in Tab. 5.

$$\text{FLOPs}_{\text{(per step)}} = 2 \times \text{FLOPs}_{\text{(per forward pass)}} \times \text{Batch size}$$
$$\text{FLOPs}_{\text{(per model)}} = \text{FLOPs}_{\text{(per step)}} \times \text{PPO epochs} \times (\text{\# of iterations})$$
$$\text{FLOPs}_{\text{(per morphology)}} = \text{FLOPs}_{\text{(per model)}} / (\text{\# of searched morphologies}) \qquad (5)$$

DERL employs a single-agent MLP policy, resulting in a per-model training compute of $2 \times 31.9\text{k} \times 512 \times 4 \times 1220 = \underline{159}\text{B}$ FLOPs. In contrast, LOKI uses a multi-agent Transformer policy, which incurs higher compute per forward pass, as well as larger batch sizes and more training epochs. However, this higher cost is offset by the fact that each trained model serves a large number of morphologies. As a result, the total training compute per searched morphology is amortized and given by $\frac{2 \times 79.5M \times 5120 \times 8 \times 1220}{78{,}000} \approx \underline{102}\text{B}$ FLOPs. Despite the higher overall training cost of the Transformer policy compared to the MLP, its shared usage across a large number of morphologies leads to more sample-efficient training, resulting in approximately 40% lower compute cost per morphology.

Table 5: Comparison of total **FLOPs** required for MLP and Transformer policy architectures.

| Model | FLOPs (per forward pass) | Batch size | PPO Epochs | # of evaluated morphologies (per model) |
|---|---|---|---|---|
| MLP | 31.9K | 512 | 4 | 1 |
| Transformer | 79.5M | 5120 | 8 | 78,000 |

## G MAP-Elites

We provide the algorithm for the Map-Elites [23, 49] baseline in Alg. 2.

---

**Algorithm 2** MAP-ELITES

---

1: **Input:**
    $\mathcal{M}$: MAP-Elites repertoire for agent design
    $\mathcal{U}$: UNIMAL morphology space
    $\boldsymbol{g}$: Cluster classifier based on morphology parameters
    $N_{\text{train}}$: Total number of agents to train
    $M$: Number of offspring per generation
    $S$: Number of interaction steps for training each MLP policy
2:  // Initialization
3: Randomly sample $M$ initial morphologies $\{u_i^0\}_{i=1}^{M} \subset \mathcal{U}$
4: Train single-agent MLP policies $\{\pi_i^0\}_{i=1}^{M}$ on their respective morphologies for $S$ steps
5: Evaluate fitness $\{f_i^0\}_{i=1}^{M}$ via one episode roll-out per trained policy
6: Insert $\{u_i^0\}_{i=1}^{M}$ into $\mathcal{M}$ using fitness $\{f_i^0\}_{i=1}^{M}$ and cluster assignments $\{\boldsymbol{g}(u_i^0)\}_{i=1}^{M}$
7: **while** $|\mathcal{M}| < N_{\text{train}}$ **do**
8:    // Reproduction via mutation
9:    Sample $M$ elite morphologies $\{u_i^j\}_{i=1}^{M}$ from $\mathcal{M}$ without replacement
10:   Apply random mutation to produce $M$ offspring morphologies $\{\tilde{u}_i^j\}_{i=1}^{M}$
11:   // Train and evaluate new morphologies
12:   Train MLP policies $\{\pi_i^j\}_{i=1}^{M}$ for $S$ steps
13:   Evaluate fitness $\{f_i^j\}_{i=1}^{M}$ via one episode per policy
14:   Insert $\{\tilde{u}_i^j\}_{i=1}^{M}$ into $\mathcal{M}$ using fitness $\{f_i^j\}_{i=1}^{M}$ and clusters $\{\boldsymbol{g}(\tilde{u}_i^j)\}_{i=1}^{M}$
15: **end while**

---

# H    Implementation Details

Detailed hyperparameters are provided in Tab. 6 and Tab. 7.

Table 6: **Hyperparameters of LOKI.**

| | Name | Value |
|---|---|---|
| **Stochastic Multi-Design Evolution** | # of samples for K-means | $5 \times 10^6$ |
| | # of clusters $N_c$ | 10, 20, 40 |
| | $N_{\text{iter}}$ | 1220 |
| | $f_{\text{diff}}$ | 2 |
| | $N_w$ | 20 |
| | $N_{\text{filter}}$ | 2 |
| | $N_{\text{sample}}$ | 128 |
| | # of parallel environments | 32 |
| | Total interactions | $10^8$ |
| | Timesteps per PPO rollout | 2560 |
| | PPO epochs | 8 |
| | Training episode length $l_{\text{train}}$ | 1000 |
| | Evaluation episode length $l_{\text{eval}}$ | 200 |
| **Multi-Design Transformer Policy** | # of heads | 2 |
| | # of layers | 5 |
| | Batch size | 5120 |
| | Feedforward dimension | 1024 |
| | Dropout | 0.0 |
| | Initialization range for embedding | [-0.1, 0.1] |
| | Initialization range for decoder | [-0.01, 0.01] |
| | Limb embedding size | 128 |
| | Joint embedding size | 128 |
| **Morphology VAE** | Continuous feature embedding size | 32 |
| | Depth feature embedding size | 32 |
| | Latent dimension | 32 |
| | # of heads | 4 |
| | Feed-forward network hidden dimension | 256 |
| | # of layers | 4 |
| | Opitmizer | Adam |
| | Initial learning rate | $10^{-4}$ |
| | Weight decay | $10^{-5}$ |
| | Learning rate scheduler | ReduceLROnPlateau [65] |
| | Learning rate reduction factor | 0.95 |
| | Learning rate reduction patience | 10 |
| | # of epochs | 200 |
| | Batch size | 4096 |
| | $[\beta_{\min}, \beta_{\max}]$ | $[10^{-5}, 10^{-2}]$ |

Table 7: **Hyperparameters of MAP-ELITES.**

| Name | Value |
|---|---|
| # of samples for K-means | $5 \times 10^6$ |
| # Clusters | 40 |
| $N_{\text{train}}$ | 4000 |
| $M$ | 100 |
| $S$ | $5 \times 10^6$ |

# I    Societal Impacts

**Positive impacts:** Our work aims to challenge prevailing assumptions about optimal robot morphology, potentially reshaping how robotic design is approached. By promoting diversity and functionality

beyond conventional forms, LOKI encourages exploration of unconventional yet effective morphologies. We believe this contributes toward a more inclusive and biologically inspired understanding of embodiment, potentially serving as a bridge between AI, robotics, and the natural sciences.

**Negative impacts:** A potential concern is that the ability to autonomously generate a large number of novel and capable morphologies could be misused in contexts that prioritize performance over safety. For instance, this approach could enable rapid prototyping of morphologies for autonomous systems without adequate human oversight, increasing the risk of deploying untested designs in sensitive environments.

