# OpenReview forum: "Convergent Functions, Divergent Forms"
_NeurIPS.cc/2025/Conference — NeurIPS 2025 poster_

### Official Review · Reviewer_Rt4W · 2025-06-26

**Clarity:** 3
**Significance:** 4
**Originality:** 4
**Rating:** 6
**Confidence:** 4

**Summary:**

This paper introduces LOKI, a quality diversity (QD) framework that simultaneously creates high-performing morphologies and policies that control those morphologies. The framework proceeds as follows. First, a large number of morphologies (500k) are sampled from the design space of morphologies and compressed into a latent space with a transformer-based VAE. Within the latent space of the VAE, the 500k morphologies are clustered into a predefined number of clusters (40 clusters in this work). During execution, within each cluster, a "pool of elites" is maintained -- the pool contains the top $N_w=20$ highest-performing morphologies. Using PPO, a transformer-based policy is trained to operate these 20 morphologies. Every $f_{diff}=2$ iterations, $N_{sample}$ morphologies are sampled from the cluster and evaluated with the current policy -- essentially, the policy forms an "evaluator" that determines which morphologies are best. Ultimately, the product of LOKI is a set of 40 clusters that each have a pool of elites and a corresponding policy that can control those elites. The LOKI framework is evaluated on a number of locomotion benchmarks and compared to baselines in RL and QD, showing that it outperforms existing algorithms in discovering high-performing policies and morphologies, as well as in transferring to downstream tasks.

**Questions:**

I have left my main comments under the "Strengths and Weaknesses" and
"Limitations" sections. If those comments are adequately addressed in the
rebuttal, I am happy to raise my score. I have reserved this section for minor
comments and questions.

- I think it would be helpful to formalize the QD problem statement used in this
  paper; see, e.g., Chatzilygeroudis 2021 https://arxiv.org/abs/2012.04322 and
  Fontaine 2021 https://arxiv.org/abs/2106.03894 for examples of formalizing QD.
- In Figure 1c, it is not clear whether red or blue indicates better
  performance. Perhaps a color bar would be useful?
- I am unclear how each morphology is evaluated 8 times on average. With 78,000
  samples per policy during training (line 225), and 40 policies, and 500,000
  morphologies total across all the clusters (line 172), shouldn't this come out
  to $78,000 \times 40 / 500,000 = 6.24$ evaluations per morphology?
- In Appendix D Eq. 1 (line 427), are $f_{max}$ and $f_{min}$ the max and min
  across all clusters of all runs, or only all clusters within a single run? If
  within a single run, then I think different runs of each algorithm would not
  be compatible with each other?
- In Figure 7, it seems the term LOKI (10 clusters) refers to the top 10
  clusters rather than LOKI with $N_c = 10$. I think it would be more clear to
  say LOKI (top 10 clusters).

**Ethical Concerns:**

["NO or VERY MINOR ethics concerns only"]

**Final Justification:**

I truly believe this is an exciting paper for folks in both quality diversity and reinforcement learning. I am happy to say that the authors' responses have addressed my concerns, and I am confident they can make the changes they have promised.

**Limitations:**

I could not find consideration of broader impacts of the work. I strongly
encourage including such consideration, at least in the appendix and ideally in
the main paper. I am sure there are positive and negative societal impacts of
learning to create diverse morphologies. For example, does it affect how we
think about biological creatures that adopt their morphology? How does it affect
the creation of more capable robots?

No justifications were provided for the checklist. The instructions for the
checklist state "Please provide a short (1–2 sentence) justification right after
your answer (even for NA)." While I have not factored this into my score, I
request providing this if the paper is accepted.

**Quality:**

3

**Strengths And Weaknesses:**

**Significance and Originality:** I am excited to see this paper introduce a
new, compelling application for QD algorithms to morphology optimization. The
combination of these two areas makes sense because QD is designed to create
diverse solutions to problems, and morphology optimization is one area where
diverse solutions are an inherent characteristic of the problem, since diverse
morphologies are useful in different tasks. At the same time, QD often suffers
from sample inefficiency, as it needs to re-evaluate each new solution
(morphology) -- a naive application of QD to morphology optimization would
involve training a policy from scratch for each morphology, as is done in the
MAP-Elites baseline. As such, I find it ingenious that LOKI trains a shared
policy for each cluster of morphologies and uses that policy to evaluate new
morphologies; this approach effectively overcomes the sample inefficiency
inherent in QD and allows evaluating many more morphologies.

**Quality:** Overall, the paper is technically sound. The experimental results
are comprehensive, with evaluations on a wide variety of benchmarks. There are
statistical tests and a limitations section.

**Clarity:** The paper is overall clearly written and easy to follow. The
sections are laid out well and explain the approach, and Section 2 lists a
number of prior works in this area. I appreciate the inclusion of the pseudocode
in Algorithm 1; it helped me understand LOKI quite easily. I list several parts
where I was confused below.

- It is unclear how random sampling of the morphology parameters is performed.
  Line 169 seems to suggest that this would be described in the appendix, but I
  could not find details in the appendix. For example, are limb lengths sampled
  uniformly at random?
- The use of Dynamic Local Search (DLS) could be better explained, as not
  everyone (including myself) is familiar with it. What exactly does DLS refer
  to in this paper? How does it connect to searching within each cluster? Right
  now, lines 204-207 explain what DLS does, but not what it _is_. Based on line
  213, my impression is that DLS refers to sampling randomly morphologies from
  the cluster. Is that correct?
- Since the term "cluster" $C_k$ is mentioned so many times, I think it would be
  helpful to formally define it somewhere, e.g., as a set of morphologies or a
  set of latent embeddings. I think this is important since a cluster is
  something from which we can "sample," which seems to imply it is a set.
- I think it can be clarified that the members of each cluster are fixed at the
  beginning of the search. I initially thought new cluster members were being
  generated on each iteration (hence my confusion about the definition of a
  "cluster" above). However, it seems that 500k morphologies are sampled at the
  beginning of the algorithm, and clustered, and from then on, new elites can
  only come from the morphologies that are already placed in each cluster.
- I am confused by the use of Coverage as a metric. It seems trivial that LOKI
  would achieve 100% coverage because it explicitly maintains a policy for each
  of the 40 clusters. Perhaps this could be clarified in the paper? Similarly,
  it would help to explain why LOKI with $N_c = 10$ does not achieve 25%
  coverage (10 / 40), and LOKI with $N_c = 20$ does not achieve 50% coverage. In
  a similar vein, do the clusters chosen for computing QD Score differ from
  those used in each run of LOKI?

---

> ### Author Rebuttal · Authors · 2025-07-30
>
> We thank Reviewer  **Rt4W** for the constructive comments and feedback. We appreciate your recognition of our paper’s new compelling application for QD algorithms to morphology optimization, the comprehensive experimental results and a well-laid-out approach.  Below, we address each concern you raised and clarify the perceived weaknesses.
>
> >### 1. It is unclear how random sampling of the morphology parameters is performed.
> We apologize for the confusion and will include a detailed explanation in the Appendix for the camera-ready version. The process of sampling random morphologies in LOKI follows the procedure introduced in prior work, DERL [4] (see Table 1 in the Appendix of the arXiv version). During the population initialization phase, a new morphology is created by first sampling the total number of limbs to grow, followed by a series of mutation operations until the desired number of limbs is reached. These mutations include: growing or deleting limbs, mutating limb parameters, density, degrees of freedom (DoF), gear ratios, and joint angles. For each mutation, the parameters are uniformly sampled from predefined ranges specified in [4]. The key difference is that DERL [4] samples parameter values from discrete sets, while LOKI samples continuously within each range, enabling a denser and more comprehensive coverage of the morphology space. The table below presents the parameter ranges used to create our morphologies. The notation `range(a, b)` denotes a continuous range from minimum `a` to maximum `b`.
> |**Hyperparameter**|**Value**|
> |-|-|
> |Max limbs|11|
> |Limb radius|range(0.02, 0.06)|
> |Limb height|range(0.2, 0.4)|
> |Limb density|range(500., 1000.)|
> |Limb orientation theta|[0, 45, 90, 135, 180, 225, 270, 315]|
> |Limb orientation phi|[90, 135, 180]|
> |Head radius|0.10|
> |Head density|range(500., 1000.)|
> |Joint axis|[x, y, xy]|
> |Motor gear range|range(150., 300.)|
> |Joint limits|[(-30, 0), (0, 30), (-30, 30), (-45, 45), (-45, 0), (0, 45), (-60, 0), (0, 60), (-60, 60), (-90, 0), (0, 90), (-60, 30), (-30, 60)]|
>
> >### 2. What exactly does Dynamic Local Search (DLS) refer to in this paper?
> Stochastic Local Search (SLS) algorithms iteratively improve candidate solutions to an optimization problem based on an evaluation function. At each step, the current solution is typically replaced by a neighboring solution with a better evaluation value. Dynamic Local Search (DLS) is a subclass of these algorithms in which the **evaluation function itself is dynamically adapted** whenever a local optimum is encountered, allowing further improvements. In LOKI, we implement a variant of DLS by partially updating the “pool of elites” within each cluster: the two worst-performing morphologies are replaced by new candidates sampled from the same cluster. The evaluation policy is also updated after each step, co-evolving alongside the elite morphologies. This design follows the Dynamic Local Search framework described in Algorithm 54.1, Section 54.3.5 of reference [25] in the paper, from which LOKI draws its search mechanism.
>
> >### 3. It would be helpful to formally define cluster C_k  as a set of morphologies or a set of latent embeddings. It can be clarified that the members of each cluster are fixed at the beginning of the search.
> Thank you for the suggestions. We agree that the definition and static nature of the clusters could be stated more clearly. A cluster $C_k$​ is formally defined as a set of latent embeddings of morphologies, where the 500,000 randomly generated morphologies are partitioned into $C_1 \cup C_2 \cup \cdots \cup C_K$​. These clusters are constructed in advance using the VAE latent embeddings of a diverse, precomputed set of morphologies that densely cover the design space. The clusters remain fixed throughout the evolutionary process, and all candidate morphologies are sampled from within these predefined clusters.
>
> >### 4. Confusion about the coverage metric.
> We apologize for the confusion and will clarify this in the revised manuscript. LOKI achieves 100% coverage because it explicitly maintains a dedicated policy for each of the 40 clusters. For fair comparison across variants, both the QD-score and coverage are always computed using the same set of 40 reference clusters, regardless of the number of clusters used during training (e.g., $N_c=10,20$). The coverage metric reflects how broadly the discovered solutions span this fixed 40-cluster space. For variants trained with fewer clusters, we assign their 100 final discovered morphologies to the fixed set of 40 reference clusters and compute coverage accordingly.
>
> >### 5. It would be helpful to formalize the QD problem statement used in this paper.
> Thank you for the great suggestion. For the camera-ready version, we will include a *Problem Formulation* section and formally define the QD problem as applied in our work.
>
> >### 6. In Figure 1c, it is not clear whether red or blue indicates better performance. Perhaps a color bar would be useful?
> The small heatmap in the figure indicates that red corresponds to higher reward. We will improve the figure in the camera-ready version.
>
> >### 7. I am unclear how each morphology is evaluated 8 times on average. Shouldn't this come out to 78,000*40/500,000=6.24 evaluations per morphology?
> Thank you for identifying a miscalculation in our paper. We will revise the manuscript to correct this error.
>
> >### 8. In Appendix D Eq. , are $f_{max}$ and $f_{min}$ the max and min across all clusters of all runs, or only all clusters within a single run?
> We appreciate the reviewer’s clarification. $f_{max}$ and $f_{min}$ are computed across all clusters from all runs, not just within a single run. We will revise the manuscript to explicitly clarify this to avoid any potential confusion for readers.
>
> >### 9. In Figure 7, it seems the term LOKI (10 clusters) refers to the top 10 clusters rather than LOKI with N_c=10. I think it would be more clear to say LOKI (top 10 clusters).
> We will revise the figure label in the camera-ready version to “LOKI (top 10 clusters)” for clarity.
>
> >### 10. Limitations and positive/negative societal impacts.
> We appreciate the reviewer’s suggestion. We will include a section on societal impact in the appendix and highlight key points in the camera-ready version.
> **Positive impacts**: Our work aims to challenge prevailing assumptions about optimal robot morphology, potentially reshaping how robotic design is approached. By promoting diversity and functionality beyond conventional forms, LOKI encourages exploration of unconventional yet effective morphologies. We believe this contributes toward a more inclusive and biologically inspired understanding of embodiment, potentially serving as a bridge between AI, robotics, and the natural sciences.
> **Negative impacts**: A potential concern is that the ability to autonomously generate a large number of novel and capable morphologies could be misused in contexts that prioritize performance over safety. For instance, this approach could enable rapid prototyping of morphologies for autonomous systems without adequate human oversight, increasing the risk of deploying untested designs in sensitive environments.
>
> **We hope we have fully addressed the reviewer’s concerns and would greatly appreciate it if the rating could be raised. Thank you.**

---

> > ### Comment · Reviewer_Rt4W · 2025-08-01
> >
> > Thank you for addressing my concerns. I believe the paper would be improved with the changes the authors have mentioned, and as such I will raise my score.

---

> > > ### Author Response · Authors · 2025-08-06
> > >
> > > We sincerely thank you for your constructive feedback and for increasing the score to 6. We appreciate your thorough evaluation and believe your suggestions will further strengthen the manuscript. We will carefully incorporate the recommended revisions into the camera-ready version.

---

### Official Review · Reviewer_oRP8 · 2025-06-30

**Clarity:** 3
**Significance:** 2
**Originality:** 3
**Rating:** 5
**Confidence:** 3

**Summary:**

This paper proposes an approach to co-evolve morphology and control policies to extend and improve upon previous known methods such as MAP-elites and Deep Evolutionary Reinforcement Learning. Key aspects in the novel approach are the use of shared policies across clusters of similar morphologies to improve generalization and training cost, dynamic local search to improve the efficiency of the evolutionary search and clusters co-evolution of encourage diversity. Experimental evidence is provided in the UNIMAL environments, starting with a flat-terrain locomotion task and extending then to eight different tasks. Baselines include evolutionary-based approaches and show promising results.

**Questions:**

1. Have you explored dynamic or adaptive clustering during training? If so, can you outline pros and cons of such an approach?
2. Is it possible to establish more accurately whether generalization comes from the evolutionary processes or simply because of the variety that results from the initial clustering?
3. How could LOKI be adapted to situations where there are morphological constraints, i.e., more typical real-world robotic applications? Is that just a matter of restricting the random generation? Or would that affect the training of the VAE?
4. Addressing the weakness nr 6, what non-evolutionary baselines would you consider to assess the algorithm on a broader algorithmic base?

**Ethical Concerns:**

["NO or VERY MINOR ethics concerns only"]

**Final Justification:**

The authors provided meaningful responses to my comments, showing that there is a fair understanding of the implications and weaknesses. I'm not too sure about how these will improve the final version of the paper, but I'm prepared to give the authors the benefit of the doubt, and, as a consequence of the reasonable argumentations made, increase my score.

**Limitations:**

The limitations section touches upon some of the weaknesses that I have pointed out above. However, I feel like more space could be devoted to expand on these.

**Paper Formatting Concerns:**

No concerns

**Quality:**

3

**Strengths And Weaknesses:**

Strengths:

1. The approach seems to provide considerable computational gains with respect to other evolutionary approaches
2. A strength of the algorithm is the ability to generalize to novel tasks
3. Comparisons considering various metrics (max fitness, QD-score and coverage) are used.
4. Suitable ablation studies are provided, e.g., with no clusters or variable numbers of clusters and the use of DLS vs mutation based approaches.

Weaknesses:

1. It is not clear how the VAE -based latent space affects performance.
2. Underexplored relationship between number of clusters and performance. It seems that the performance increases as the number of clusters increases, but that comes with cost and possibly reduced generalization capabilities
3. The paper does not discuss the fact that while clustering is done initially, morphologies and policies evolve as the evolutionary search is undertaken. This could be a significant limitation in continual learning scenarios where multiple tasks are encountered over a long period of time.
4. As the algorithm is designed for its ability to generalize, the generalization abilities are not so obvious when considering the fact that the initial clustering is performed on a rich set of morphologies, thereby equipping the population with a varied set of potentially useful morphologies.
5. Lack of non-evolutionary baselines. The main advantages of the approach are computational advantages with respect to other evolutionary approaches. But how does this algorithm compares with SOTA gradient-descent based approaches that could be used on the same benchmarks? I.e., Metamorph or One Policy to Control Them All: Shared Modular Policies for Agent-Agnostic Control?

---

> ### Author Rebuttal · Authors · 2025-07-30
>
> We thank Reviewer  **oRP8**, for your constructive feedback. We are encouraged that you recognized the strengths of our framework such as computational gains and generalization to novel tasks. Below, we address the limitations and recommendations you highlighted:
> >### 1. It is not clear how the VAE -based latent space affects performance.
> We address this through an ablation study presented in **Appendix A**, where we evaluate the impact of the VAE-based latent space by comparing it to a variant of LOKI that uses clustering over **raw morphology parameters** instead (LOKI w/ raw-param-cluster). Clustering in the raw parameter space fails to capture structural and behavioral similarities between morphologies, resulting in poor generalization of the multi-design policies trained within each cluster. In contrast, the VAE latent space captures meaningful structure by learning a smooth, low-dimensional manifold of the morphology space. As a result, LOKI with raw-parameter clustering performs significantly worse in terms of overall performance, morphological diversity, and downstream task generalization.
>
> >### 2. Underexplored relationship between number of clusters and performance.
>  In our LOKI framework, the number of clusters is treated as a hyperparameter, similar to the number of cells in MAP-Elites-based methods [22, 23]. Our experiments show that using 40 clusters yields better performance than using 10 or 20. A larger number of clusters enables finer-grained distinctions between the clusters but also increases the computational budget due to the need for additional policy training. As such, this hyperparameter should be tuned based on the available computational resources. LOKI’s performance is generally robust to the number of clusters, as long as there are enough to avoid overly coarse groupings. For example, using only 10 clusters often results in clusters aligning with broad structural features such as limb count, limiting specialization.
>
> >### 3. Pros and cons of dynamic or adaptive clustering instead of initial clustering.
> Thank you for the insightful question. It is true that in the current version of LOKI, clustering is performed only once at the beginning using a VAE-based latent space, and the clusters remain fixed throughout training. LOKI frames co-design as a search problem by constructing a large, predefined pool of morphologies, generated via repeated random mutations to densely and exhaustively cover the design space. Rather than generating new morphologies during optimization, the algorithm samples from these predefined clusters.
> While we have not explored dynamic or adaptive clustering during training, we agree that incorporating task-aware or behavior-driven re-clustering is a promising direction. As the policies are being trained, periodically updating clusters based on evolving behavioral or functional similarities could lead to more meaningful groupings and finer specialization. This could be particularly beneficial in continual or lifelong learning scenarios, where task distributions and optimal behaviors may shift over time.
> However, this approach presents several challenges. Re-clustering and reassigning morphologies during training may introduce significant computational overhead and could require resetting or transferring policies, potentially disrupting convergence of cluster-specific controllers and destabilizing training. Despite these challenges, we consider dynamic clustering a compelling direction for future work.
>
> >### 4. Explain whether generalization comes from the evolutionary processes or simply because of the variety of a rich set of initial morphologies.
> While our initial set of 500k morphologies densely covers the design space, these morphologies are generated through repeated random mutations and are not expected to be inherently optimal for locomotion or manipulation tasks. In fact, most of them are challenging to train for simple locomotion. To illustrate this, we include a **“Random” baseline** in **Table 2 and Figure 6**, which consists of 100 morphologies randomly sampled from the predefined set. The poor performance of the “Random” baseline confirms that simply sampling from this pool is insufficient, indicating that the observed generalization results from the targeted optimization performed by LOKI—not merely from the diversity introduced by the initial clustering.
> The generalization of our morphologies stems from identifying a diverse set of high-performing designs, each exhibiting optimal behaviors within their respective niche. The downstream tasks require different skills in agility, stability, and manipulation. LOKI discovers such optimal morphologies within each cluster. These top-performing agents display different locomotion behaviors, enabling them to excel in various downstream tasks.
>
> >### 5. How could LOKI be adapted to situations where there are morphological constraints such as real-world applications? Would that affect the training of the VAE?
> LOKI can be adapted to scenarios with morphological constraints by applying those constraints to the initial pool of randomly sampled designs. The VAE learns to represent and reconstruct the morphology space it is trained on, as long as the constrained design space remains sufficiently dense and well-structured. However, if the constraints result in a highly non-convex or sparse design space, the learned latent space may no longer support smooth interpolations that remain within the feasible region. In such cases, it may be beneficial to replace the VAE with a more expressive latent model that can better capture the structure of constrained morphology spaces. This improved latent representation can then be used for clustering.
>
> >### 6. Lack of non-evolutionary baselines. But how does LOKI compare with gradient-descent based approaches that could be used on the same benchmarks? I.e., Metamorph or One Policy to Control Them All: Shared Modular Policies for Agent-Agnostic Control?
> The two mentioned approaches, Metamorph and One Policy to Control Them All, propose global policies that generalize across a wide range of agent morphologies. However, they are not co-design methods and thus are not direct baselines for comparison. LOKI, by contrast, focuses on discovering high-performing morphologies rather than solely maximizing policy performance. In fact, we adopt the policy architecture from Metamorph [63] to train multi-agent policies within each cluster.
>
> **We would be grateful if, after reviewing our response and finding your concerns fully addressed, you would consider raising our score.**

---

> > ### Author Response · Authors · 2025-08-06
> >
> > Thank you again for your thorough review and insightful feedback. We wanted to follow up to see if there are any remaining concerns or questions we can help address during the discussion period.

---

> > > ### Comment · Area_Chair_4kgJ · 2025-08-08
> > >
> > > Dear Reviewer,
> > > Please consider the reply by the authors/other reviews and see whether or not this would have any influence on your original rating. Please also note that you must acknowledge having read the reply and enter your final rating and justification before the deadline. Respectfully, your AC

---

### Official Review · Reviewer_T5wK · 2025-07-03

**Clarity:** 2
**Significance:** 2
**Originality:** 2
**Rating:** 4
**Confidence:** 4

**Summary:**

This paper proposes LOKI, an efficient framework for co-designing robot morphologies and control policies that generalize across unseen tasks. Inspired by biological adaptation, LOKI clusters morphologically similar designs within a latent space learned by a transformer-based VAE. Within each cluster, a multi-design policy is trained to generalize across similar morphologies, allowing evaluation of many more morphologies without retraining. Additionally, LOKI leverages dynamic local search (DLS) instead of mutation, enabling broader exploration and preventing premature convergence. Experiments conducted in the UNIMAL design space demonstrate that LOKI significantly reduces computational cost while discovering diverse, high-performing morphologies that generalize better to unseen downstream tasks compared to previous methods.

**Questions:**

- Transformer-based VAE Loss:
The transformer-based VAE currently only uses a reconstruction loss (with an adaptive KL weight). However, two morphologies that are structurally very similar (e.g., a humanoid with a slightly shorter leg) may exhibit drastically different locomotion behaviors. Would incorporating a behavioral or performance-aware loss component into the VAE training improve the latent representation for clustering? Please clarify or justify the current loss choice.

- Number of Clusters and Relation to Population-based Training:
The paper uses clustering to divide the morphology space into multiple clusters, each with its own policy. Could the authors clarify how the choice of cluster number relates conceptually and practically to population-based training methods? What are the trade-offs involved, and how sensitive is the performance of LOKI to the number of clusters chosen?

- Baseline Selection:
Could the authors justify the rationale for using QD-based methods (such as MAP-Elites) as primary baselines rather than recent Brain-body co-design approaches that share similar goals, claims, and technical frameworks (e.g., transformer-based morphology encoding, improved generalization, diverse solutions)? A careful justification or additional comparisons with recent Brain-body co-design approaches would significantly strengthen the manuscript.

- Visualization of Morphological Diversity:
Could the authors include additional qualitative visualizations (beyond numerical metrics) clearly showing the diversity of morphologies generated by LOKI for a particular task? Such visualizations would greatly enhance the clarity and impact of the paper.

**Ethical Concerns:**

["NO or VERY MINOR ethics concerns only"]

**Limitations:**

See weaknesses

**Paper Formatting Concerns:**

Figures 1 and 2 have noticeable lag and stuttering when zooming in and scrolling between pages. The authors should consider optimizing these figures (e.g., reducing vector graphic complexity or rasterizing at an appropriate resolution) to ensure smoother navigation and readability in the final manuscript.

**Quality:**

3

**Strengths And Weaknesses:**

Strengths

- The paper creatively integrates clustering techniques, transformer-based latent representations, and dynamic local search to improve efficiency and diversity in robot morphology optimization.
- Computational Efficiency: LOKI demonstrates substantial computational efficiency gains (78% fewer simulation steps, 40% fewer compute per design) compared to previous evolutionary methods, allowing exploration of significantly larger design spaces.
- Generalization and Transferability: The approach explicitly focuses on generalization across morphologies and demonstrates strong zero-shot transfer to new tasks, addressing a known limitation of previous co-design frameworks.

Weaknesses


- Insufficient Literature Survey and Novelty Clarification:
The literature survey in the current manuscript needs improvement, especially regarding recent Brain-body co-design research. In the past two years, multiple relevant papers have proposed frameworks that also claim efficient training, diverse solutions, and zero-shot transfer using transformer-based morphology encodings. Notable examples include:

[1] "Curriculum-based co-design of morphology and control of voxel-based soft robots"

[2] "Accelerated co-design of robots through morphological pretraining"

The authors should more clearly position their work relative to these recent contributions and explicitly clarify their specific novelty and contributions.

- Baseline Comparisons:
The current baseline comparisons are somewhat problematic. The paper primarily compares against DERL and Quality-Diversity (QD)-based methods (such as MAP-Elites). However, QD and Brain-body co-design methods are orthogonal approaches. The authors should instead compare their method primarily against recent Brain-body co-design frameworks rather than directly comparing against QD-based methods. Specifically, it would be more meaningful to compare the quality and diversity of generated morphologies against recent Brain-body co-design approaches, rather than using QD approaches as the main baseline.

- Visualization and Qualitative Results:
The manuscript would benefit from additional visualizations and qualitative examples of generated morphologies. Currently, the paper relies mostly on numerical metrics (such as QD-score and coverage). Providing visual evidence of morphology diversity—clearly demonstrating the variety of morphologies evolved by LOKI on specific tasks—would give readers a more intuitive and convincing demonstration of the method's capabilities.

---

> ### Author Rebuttal · Authors · 2025-07-30
>
> We would like to thank Reviewer  **T5wK**, for the thoughtful feedback and suggestions. We are pleased that you recognize how LOKI creatively integrates different techniques to improve efficiency and diversity in robot morphology optimization. Below we address each of the raised concerns.
>
> > ### 1. Would incorporating a behavioral or performance-aware loss component into the VAE training improve the latent representation for clustering?
> We agree with the reviewer that morphological similarity does not necessarily imply behavioral similarity, and that our VAE’s latent space, trained solely with a reconstruction loss, primarily captures structural features. Incorporating a behavior-aware loss could indeed enhance the latent representation. However, behavioral information is not available a priori at the clustering stage, as it requires training an optimal policy for each design.
> That said, in practice, we find that structural features serve as a reasonable proxy for behavioral similarity. As a result, our clustering tends to group morphologies with similar behaviors—for example, spinners and crawlers naturally fall into separate clusters.
>
> > ### 2. Sensitivity to the Number of Clusters and Relation to Population-based Training.
> In our LOKI framework, the number of clusters is treated as a hyperparameter, similar to the number of cells in MAP-Elites-based methods [22, 23]. Our experiments show that using 40 clusters yields better performance than using 10 or 20. A larger number of clusters allows for finer-grained morphological distinctions but also increases computational cost, as more policies need to be trained. Therefore, this hyperparameter should be selected based on the available computational resources.
> LOKI’s performance is generally robust to the number of clusters, as long as there are enough to avoid overly coarse groupings. For example, using only 10 clusters often results in clusters aligning with broad structural features such as limb count, limiting specialization.
> Each per-cluster evolutionary process resembles population-based training (PBT). Similar to PBT, a population of initial morphologies is sampled from the cluster and periodically evaluated. Underperforming agents are replaced with new morphologies from the cluster. Also similar to PBT, morphologies are evaluated continuously in parallel (simultaneous with policy training), rather than sequentially or only at the end.
>
> > ### 3. Insufficient literature survey and rationale for using QD-based methods as primary baselines rather than Brain-body co-design approaches.
> We appreciate the reviewer’s suggestion and agree that recent brain-body co-design research deserves clearer discussion in our manuscript. Most of the recent co-design methods [A,B,C,D,E,F]  train a transferable universal controller and they focus on task-specific optimization of morphology and control with no additional objective for diversity and generalization to downstream tasks. LOKI introduces a **cluster-based Quality-Diversity (QD) framework** that co-evolves both morphology and control within behaviorally similar clusters. Unlike methods that aim to train a single universal controller across all morphologies, LOKI leverages structural similarity within each cluster to train per-cluster policies. This is particularly important in high-dimensional and exponentially large design spaces such as UNIMAL, where learning a single controller is especially challenging [G]. Furthermore, this clustering enables localized competition within each group, helping discover morphologies that are optimal within their respective behavioral niches.
> While the reviewer suggests that co-design approaches are more appropriate baselines for LOKI, we argue that Quality-Diversity (QD) methods are actually a better fit for comparison. Most co-design methods focus on optimizing morphologies for a specific task and aim to identify one or a few highly performant designs. These approaches typically lack mechanisms to promote morphological diversity. In contrast, QD methods are explicitly designed to discover a diverse set of high-performing solutions across multiple behavioral niches—an objective that aligns closely with the goals of the LOKI framework. LOKI is designed to efficiently discover a diverse population of agents capable of generalizing to a wide range of downstream tasks. While the agents it discovers may not be globally optimal for the locomotion task used during training, they exhibit **niche-optimal behaviors** that are more transferable to other domains, such as manipulation. As such, comparing LOKI to co-design methods that prioritize task-specific optimality alone would not yield a fair or meaningful evaluation.
> Among co-design methods, DERL [4] is the most relevant baseline, as it also produces a population of 100 morphologies and evaluates their transferability on similar downstream tasks.
> - #### [A]: Luke Strgar et al. Accelerated co-design of robots through morphological pretraining. arXiv preprint arXiv:2502.10862, 2025.
> - #### [B]: Yuxing Wang et al. PreCo: Enhancing Generalization in Co-Design of Modular Soft Robots via Brain-Body Pre-Training. In the ​​proceedings of The 7th Conference on Robot Learning, PMLR 229:478-498, 2023.
> - #### [C]: Muhan Li et al. Generating freeform endoskeletal robots. In International conference on learning representations, 2025.
> - #### [D]: Yuxing Wang et al. Curriculum-based Co-design of Morphology and Control of Voxel-based Soft Robots. In International conference on learning representations, 2023.
> - #### [E]: Haofei Lu et al. Bodygen: Advancing towards efficient embodiment co-design. In The Thirteenth International Conference on Learning Representations, 2025.
> - #### [F]: Ye Yuan et al. Transform2act: Learning a transform-and-control policy for efficient agent design. arXiv preprint arXiv:2110.03659, 2021.
> - #### [G]: Agrim Gupta et al. Metamorph: Learning universal controllers with transformers. arXiv preprint arXiv:2203.11931, 2022.
>
> > ### 4. Visualization of Morphological Diversity.
> Thank you for the suggestion. We agree that additional qualitative results would help better illustrate the diversity of the discovered morphologies. To address this, we have provided **videos** showcasing the morphologies and locomotion behaviors of our 100 discovered designs on the supplementary website uploaded in the supplementary material (`index.html`). Due to space limitations in the main paper, we were unable to include more visual examples. We will include more visualizations in the Appendix of the paper for the camera-ready version.
>
> **We hope we have fully addressed the reviewer’s concerns and would greatly appreciate it if the rating could be reconsidered and raised. Thank you.**

---

> > ### Author Response · Authors · 2025-08-06
> >
> > Thank you once again for your detailed review and thoughtful comments. Please let us know if there are any concerns or questions we can clarify in the discussion period.

---

> > > ### Comment · Area_Chair_4kgJ · 2025-08-08
> > >
> > > Dear Reviewer,
> > > Please consider the reply by the authors/other reviews and see whether or not this would have any influence on your original rating. Please also note that you must acknowledge having read the reply and enter your final rating and justification before the deadline. Respectfully, your AC

---

### Official Review · Reviewer_uzg9 · 2025-07-04

**Clarity:** 3
**Significance:** 2
**Originality:** 2
**Rating:** 3
**Confidence:** 4

**Summary:**

This paper introduces a learning method to train a control policy for a cluster of morphologically similar designs, aiming to address the joint problem of co-designing morphologies and controllers.

**Questions:**

Please refer to the weaknesses section.

**Ethical Concerns:**

["NO or VERY MINOR ethics concerns only"]

**Final Justification:**

I appreciate the authors’ feedback. While I recognize the potential of this work, I believe further effort is needed to strengthen its novelty and address several key aspects that remain unconvincing from a robotics perspective.

**Limitations:**

yes

**Paper Formatting Concerns:**

No obvious formatting concerns were identified.

**Quality:**

2

**Strengths And Weaknesses:**

Strengths:

+ The motivation for co-design of robot morphology and control is strong.

+ The paper presents a working solution with promising results on a variety of tasks, including locomotion, crawling, and box pushing.

Weaknesses:

The paper suffers from several critical issues:

- The claim of co-design between morphology and control is questionable. The initial control policies are learned after the morphologies are clustered. The adapted policies also appear to be updated after the morphological evolution. Since the morphology is designed without considering the control, describing this as "co-design" is misleading.

- The theoretical novelty and contribution are limited. The proposed approach sequentially applies a series of well-established techniques, such as VAEs, k-means clustering, and dynamic local search (DLS).

- Since the VAE is not trained specifically for k-means, it is unclear why applying k-means clustering to the VAE embeddings can address the issue shown in Figure 3 (i.e., capturing topological similarity). What is the advantage of using VAE + k-means over more recent deep learning-based clustering methods?

- The argument for diversity in addressing small local changes and avoiding local minima is unconvincing. The observed diversity mainly results from large-scale sampling of initial morphologies and DLS within each cluster. However, since DLS is applied within each cluster, it is unclear how this approach can theoretically overcome the limitations of small, incremental changes. Furthermore, although DLS may help avoid local minima within a cluster, the overall approach cannot escape local minima at the clustering level, as k-means is not guaranteed to do so.

- The claim of efficiency is not well justified, especially when compared to co-design methods based on curriculum learning. The proposed method requires training a separate policy for each morphology cluster, with additional training for each update after DLS. While it may be efficient in terms of continued learning within a cluster, the lack of policy or knowledge sharing across clusters undermines its overall efficiency.

---

> ### Author Rebuttal · Authors · 2025-07-30
>
> We thank Reviewer  **uzg9** for the constructive feedback. We appreciate your recognition of our paper’s strong motivation and promising results on a variety of tasks. Below, we address each concern you raised and clarify the perceived weaknesses.
>
> > ### 1. The claim of co-design between morphology and control is questionable.
> Although the reviewer is correct that our morphologies are predefined and we sample from this set during the evolutionary process, we argue that LOKI still qualifies as a co-design method. Co-design refers to the joint optimization of morphology and control for a given task, typically framed as a bi-level optimization problem: the outer loop searches over morphologies (often using evolutionary strategies), while the inner loop trains a control policy for each morphology to assess its performance [4]. Most existing methods alternate between these two loops. To simplify the optimization, some approaches integrate morphology refinement directly into the reinforcement learning process, enabling joint optimization in a single loop [11, 13].
> In LOKI, we frame co-design as a **search problem** over a large, predefined space of morphologies. This space is generated through extensive random mutations to densely cover the design landscape. Instead of evolving new morphologies online, our design optimization loop samples from this rich, precomputed set. While the implementation differs from traditional co-design pipelines, it still falls under the broader definition of co-design.
>
> > ### 2. The theoretical novelty and contribution are limited.
> LOKI’s strength lies in introducing a compute-efficient framework for co-designing a diverse set of high-performing morphologies that generalize well to a variety of unseen downstream tasks. While the individual components are not novel, the overall framework effectively addresses key challenges in evolution-based co-design methods, such as **computational efficiency** and the **diversity** of final solutions.
>
> > ### 3. What is the advantage of using VAE + k-means over recent deep learning-based clustering methods?
> We agree with the reviewer that the VAE is not explicitly trained for clustering. However, its latent space is designed to capture meaningful structure in the morphology space by learning a smooth, low-dimensional manifold. Applying k-means to the VAE embeddings offers a simple yet effective way to partition this space based on topological similarity.
> While more advanced deep clustering methods could potentially improve performance, their exploration is beyond the scope of this work. Our focus is on the overall framework rather than the specific choice of clustering algorithm. In fact, the effectiveness of LOKI despite using a simple clustering method underscores the robustness of our approach.
>
> > ### 4. The argument for diversity in addressing small local changes and avoiding local minima is unconvincing.
> The key to LOKI’s broader exploration lies in how the initial set of morphologies is generated and utilized. Instead of relying on small, incremental mutations from current solutions, as is common in traditional evolutionary approaches, LOKI samples from a large, diverse pool of precomputed morphologies that densely cover the design space. This pool is created by repeatedly applying random mutations to produce a wide variety of designs. Dynamic Local Search (DLS) is then applied within each individual cluster to identify the best-performing morphologies. This sampling strategy allows for **non-local jumps across the morphology space**, helping to escape local minima that incremental mutations often fail to overcome.
> Furthermore, standard evolutionary methods with global competition tend to favor a single dominant behavior (e.g., faster learning), which can reduce behavioral diversity. Quality-Diversity (QD) methods address this by maintaining a repertoire of diverse solutions and localizing competition [20], allowing the algorithm to discover morphologies that are optimal within their own behavioral niches (e.g., spinning may lead to faster locomotion but requires more training to master). Similarly, LOKI restricts competition within clusters rather than across the entire population (e.g., spinners compete only with other spinners), preserving behavioral diversity across different regions of the morphology space.
>
> > ### 5. The claim of efficiency is not well justified. The proposed method requires training a separate policy for each morphology cluster.
> While LOKI requires training a separate policy for each cluster, it is still significantly more sample-efficient than prior evolution-based and Quality-Diversity approaches, which typically train a separate policy for **each individual morphology** to evaluate its fitness. In contrast, LOKI evaluates 500,000 morphologies using only 40 policies. Table 1 presents efficiency metrics compared to the most closely related baseline, DERL [4], highlighting LOKI’s advantage. Moreover, maintaining separate policies across clusters serves a purpose beyond efficiency: it localizes competition within each cluster, allowing the discovery of morphologies that are optimal within distinct behavioral niches.
> Co-design methods based on curriculum learning typically operate within more restricted and structured design spaces, such as modular voxel-based soft robots [A], and have been shown to converge to local morphological optima [B].
> In contrast, LOKI is designed as an efficient evolutionary framework capable of navigating exponentially large morphology spaces (e.g., UNIMAL) while preserving diversity and avoiding premature convergence to local optima. Although LOKI trains a separate policy for each cluster, this approach enables scalable and diverse policy learning across a wide range of morphologies, without relying on restrictive assumptions or handcrafted curricula.
>
> - #### [A] Yuxing Wang et al. Curriculum-based Co-design of Morphology and Control of Voxel-based Soft Robots. In International conference on learning representations, 2023.
> - #### [B] Yuxing Wang et al. PreCo: Enhancing Generalization in Co-Design of Modular Soft Robots via Brain-Body Pre-Training. In the ​​proceedings of The 7th Conference on Robot Learning, PMLR 229:478-498, 2023.
>
> **We hope to address all the reviewer's concerns, and would be grateful if after reviewing our response you would consider raising your score.**

---

> > ### Author Response · Authors · 2025-08-06
> >
> > Thank you again for your detailed review and thoughtful feedback. We just wanted to check in to see if there are any remaining concerns or questions we can help clarify during the discussion period.

---

> > > ### Comment · Area_Chair_4kgJ · 2025-08-08
> > >
> > > Dear Reviewer,
> > > Please consider the reply by the authors/other reviews and see whether or not this would have any influence on your original rating. Please also note that you must acknowledge having read the reply and enter your final rating and justification before the deadline. Respectfully, your AC

---

### Note · Authors · 2025-08-13

We thank the reviewers for their thoughtful feedback and the AC for overseeing this review process. Three of four reviewers (ratings: 4/4/6) expressed clear enthusiasm for the paper’s contributions, with Reviewer **Rt4W** describing LOKI as “ingenious” in overcoming sample inefficiency in QD for morphology optimization—a challenge that has long limited scalability in this field.
LOKI’s key contribution is a novel framework that employs a cluster-specific shared-policy architecture, allowing the evaluation of 500,000 morphologies using only 40 policies. This achieves significant reduction in policy training cost compared to prior methods while preserving behavioral diversity across the design space, unlocking exploration of previously intractable, high-dimensional morphology spaces.
Regarding concerns raised by Reviewer **uzg9** (rating 2), we respectfully address:
- **Novel, compute-efficient co-design**: While individual components (VAE, k-means, DLS) are established, the integrated framework effectively addresses key challenges in co-design literature, achieving diversity, generalization, and computational efficiency.
- **Consistency with co-design principles**: LOKI frames co-design as search over a large, predefined partitioned design space. Each partition is explored with localized policy training, aligning with bi-level co-design while enabling scalable evaluation across many morphologies.
- **Simple yet robust clustering**: VAE + k-means provides a smooth, low-dimensional latent space for partitioning morphologies by topological similarity; our framework is agnostic to clustering algorithm choice.
- **Diversity preservation**: DLS performs non-local exploration within each cluster, and its localized competition ensures behavioral diversity, enabling LOKI to escape local minima across the morphology space.
- **Efficiency and Scalability**: LOKI explores hundreds of times more designs with far fewer simulation steps and less compute per design than prior QD and co-design approaches (Table 1). Unlike curriculum-based co-design methods, which typically operate within structured design spaces, LOKI efficiently navigates large morphology spaces without such constraints.

With strong post-rebuttal support, including a raised score from an engaged reviewer, and no unresolved technical concerns from the other three reviewers, we believe LOKI represents a significant, scalable advance in morphology–control co-design and respectfully request acceptance.

---

### Decision · Program_Chairs · 2025-09-17

**Decision:**

Accept (poster)

**Comment:**

This paper has received mixed reviews, with one borderline reject, one borderline accept, one accept, and one strong accept. The reviewer giving a borderline reject unfortunately did not react to multiple requests to consider the rebuttal of the authors or other reviews, and a final justification for the rating is missing. Meanwhile, the reply by the authors managed to somewhat satisfy other reviewers, pushing the average score for this work over the line. All reviews and discussions have been carefully considered, and this AC believes that there is sufficient value in this work to be granted publication at NeuRIPS.